# Quantitative Risk Assessment in Construction Disputes Based on Machine Learning Tools

Hubert Anysz [1], Magdalena Apollo [2,*] and Beata Grzyl [2]

[1] Faculty of Civil Engineering, Warsaw University of Technology, Al. Armii Ludowej 16, 00-637 Warsaw, Poland; h.anysz@il.pw.edu.pl
[2] Faculty of Civil and Environmental Engineering, Gdańsk University of Technology, 80-233 Gdańsk, Poland; beata.grzyl@pg.edu.pl
* Correspondence: magdalena.apollo@pg.edu.pl; Tel.: +48-502-856-262

**Abstract:** A high monetary value of the construction projects is one of the reasons of frequent disputes between a general contractor (GC) and a client. A construction site is a unique, one-time, and single-product factory with many parties involved and dependent on each other. The organizational dependencies and their complexity make any fault or mistake propagate and influence the final result (delays, cost overruns). The constant will of the parties involved results in completing a construction object. The cost increase, over the expected level, may cause settlements between parties difficult and lead to disputes that often finish in a court. Such decision of taking a client to a court may influence the future relations with a client, the trademark of the GC, as well as, its finance. To ascertain the correctness of the decision of this kind, the machine learning tools as decision trees (DT) and artificial neural networks (ANN) are applied to predict the result of a dispute. The dataset of about 10 projects completed by an undisclosed contractor is analyzed. Based on that, a much bigger database is simulated for automated classifications onto the following two classes: a dispute won or lost. The accuracy of over 93% is achieved, and the reasoning based on results from DT and ANN is presented and analyzed. The novelty of the article is the usage of in-company data as the independent variables what makes the model tailored for a specific GC. Secondly, the calculation of the risk of wrong decisions based on machine learning tools predictions is introduced and discussed.

**Keywords:** artificial neural networks; association analysis; construction project; decision-supporting tools; decision trees; disputes in construction industry; risk in decision-making

## 1. Introduction

An important condition accompanying smooth and consensual implementation of a construction project is a proportional and transparent division of risks between the parties of the contract, inter alia, the consequences of any disruptions arisen from an increase in the scope of work and extension of the completion time [1]. In this context, properly structured legal and contractual solutions are crucial, as they significantly reduce the risk of conflicts between cooperating parties and, in many cases, offer the chance to solve the aforementioned problems without any court involvement [2,3].

During the construction works execution, various types of unforeseen circumstances occur, affecting the course and progress of works causing the contractor's financial loss. In practice, the factors causing serious disturbances in the contractor's operations include the necessity to introduce changes and revisions in the scope of works (so-called change orders, e.g., due to design faults), lack of access to the construction site at the planned date, the necessity to suspend works and re-mobilize, logistic problems related to supplies, organization and coordination of works conducted by several subcontractors, adverse weather conditions [4]. The numerous disruptions are particularly severe for contractors in the current context of the ongoing COVID-19 pandemic, and resulted, e.g., in delays in the delivery of materials and equipment, slowdowns in operation due to the need to comply

with existing travel restrictions, and most importantly, increased and unpredictable worker absence [5]. The occurrence of events causing a construction project cost increase and extension of its duration means that to achieve the effect agreed between the parties at the stage of signing the contract, the contractor must incur additional expenditures—higher than originally assumed. In practice, obtaining financial compensation from the ordering party (a client) is often a source of a dispute between the parties to the contract, which in many cases, may be resolved only in a court [6–8]. One can speculate that the negative effects of a continuing pandemic, in the long run, will lead to numerous conflicts between construction investment parties and increased litigation.

The results of a survey [9] conducted by Contract Advisory Services (CAS) among entities related to the construction industry in Poland (representatives of investors and contractors, dealing with the implementation of a construction project management, handling disputes between the parties to the investment process, valuation of works, including, inter alia, lawyers, engineers, management, technical, financial and administrative staff) indicate that the average value of a dispute in which the respondents participated in 2019 was PLN 52.7 million (in 2018—PLN 52.6 million). According to the forecast of experts, it will increase in the following years [9]. Disputes involving respondents in 2018 lasted an average of 29.2 months [9]. The results of the report [9] also indicate that in 2019, the construction industry in Poland saw an escalation of conflicts related to the implementation of previously concluded contracts, an increase in the number of ineffective tender proceedings and court proceedings.

According to [9], the practice-relevant causes of disputes between contracting parties are:
- increase in costs of contract execution (according to 85% of respondents),
- missing or delayed key decisions (according to 63% of respondents),
- different conditions at the construction site compared to those specified by the ordering party (a client) (according to 51% of respondents),
- deficiencies and faults in documentation for investments conducted in the "design and build" formula (44% of respondents) and " build" formula (29% of responses),
- incorrect contract administration (22%), lack of understanding of the contract by the parties and failure to meet their contractual obligations (according to 20% of respondents),
- missing or delayed payments (20%),
- disruptions caused by adverse weather conditions (17%).

According to the respondents [9], disputes arising at the stage of implementation of the contract matter do not find an amicable settlement due to:
- fear of contractual parties of being responsible for the decisions made (according to 86% of respondents),
- divergent perception of the purpose of the contract as a conflict of interests between the parties (according to 44% of respondents),
- unwillingness to take action (34%),
- ignorance and lack of qualifications of cooperating entities (19%).

Among the most popular tools for resolving disputes, respondents [9] indicated primarily the common court (71%) and the "wait-and-see" method (68%), but also negotiation (39%), mediation (3.5%) and arbitration (3.5% of respondents). Respondents considered negotiation (78%), common court (35.5%), mediation (29%), arbitration (15%) and conciliation (13.5%) to be the most effective methods of dispute resolution. The "wait-and-see" method was not considered an effective tool for resolving disputes between contracting parties (5%). It should be noted that the answers of the respondents show a clear disproportion between the methods that in practice are most often used to resolve disputes and those that are considered to be the most effective.

It may be assumed that in practice a combined strategy for resolving disputes is used. Initially, the conflicting parties try to wait out the situation (being fully aware of the ineffectiveness of this method), and in the next stage, they transfer the responsibility for resolving the dispute to a common court. This strategy is closely related to the fundamental

causes of disputes between cooperating parties, which include lack of decisiveness, the inertia to make decisions, fear of liability, and passivity to take action. An additional factor pointed out by contractors is the significant increase in the costs of construction projects and the lack of adequate valorization formulas in the contents of contracts to reflect the actual level of changes in construction output prices [10,11]. As a consequence, they result in unprofitable contracts, ineffective solutions and high social costs. Completion of an uncompleted construction contract by a contractor selected in a new tender procedure is more expensive than, for example, increasing the amount of the original contractor's remuneration or adjusting the amount of remuneration stipulated in the contract, which currently cannot be performed by the contractor due to a drastic price increase. In such circumstances, it is reasonable for the contractor to seek an independent judicial resolution of the dispute [12,13].

The results of a survey [9] conducted by CAS indicate that projects of large scope and long duration, implemented with public funds (under the provisions of the Public Procurement Law) and by large entities (e.g., government agencies) are primarily exposed to serious disputes between contracting parties. Public sector investments (primarily road, rail and energy infrastructure construction) mainly due to the high uncertainty of the contractor regarding the terms of performance of the contract matter, are considered to generate more disputes than private projects [9,14]. The scale of these investments makes the cost increase of their implementation significant. According to [9], the largest number of disputes occurs during the implementation of road infrastructure (in 90% of cases) and rail infrastructure (47% of cases). Public procurers are considered difficult business partners, characterized by a high aversion to amicable solutions. This is caused by, among other things, systemic solutions, the obligation to apply public finance discipline, and legal regulations which significantly limit flexibility, e.g., in disposing of funds and making independent decisions that consider the current circumstances of investment implementation. It may be assumed that a large number of infrastructure projects and, at the same time, the reluctance of contracting authorities to find out-of-court solutions to disputed situations will result in an increase in the number of court proceedings in the coming years.

To sum up—the practice shows that common courts and legislation fail to keep up with the frequent changes that occur in the construction process, in the area of technology, construction organization, financial and insurance instruments. These new solutions of different nature undoubtedly influence the length of proceedings, their complexity and costs connected to dispute settlement. Regardless of its original cause, a dispute where the parties involved in the project cannot find an agreement and a way to resolve the conflict within the mechanisms provided in the content of the concluded contract, is usually settled in a court. Such a solution is not beneficial for any of the parties involved—it requires a long time to wait for a court decision and generates additional costs. In this context, alternative dispute resolution tools should be taken into account, that allow to find a quicker and a relatively cheaper method to solve a conflict such as negotiations, mediation and arbitration. Moreover, in public procurement contracts, a clear asymmetry in the distribution of risks between the parties to the contract occurs. In the current situation of instability in the construction market, any changes in the project environment particularly affect in particular one of the parties to the contract. Additionally, the disproportionate distribution of the parties' responsibilities and rights in the contract give rise to difficult relationships, conflicts and, ultimately, disputes settable only in a court. In practice, the interests of the contracting authority are better protected than those of the contractor. This is caused mainly by the fact that the terms of contracts are prepared by the contracting authority, which include requirements arising under the Public Procurement Law, and they are not subject to negotiation, so contractors do not have the opportunity to introduce clauses that protect their interests. This results in a long-term litigation and the dominant position of the ordering party. Its favorable contractual provisions cause, that in many cases, the bad financial situation of the contractor is further aggravated in a court. For

this reason, a contractor's decision about legal action is fraught with additional risk and multicriteria estimation of potential gains and losses [15,16].

Decisions can be supported with multicriteria methods [17–19], but also with machine learning tools—one of possible approaches supporting this process is Bayesian statistical decision theory providing a mathematical model to make decisions in conditions of uncertainty [20]. In the context of disputes in construction industry, the authors decided however to use decision trees (DT) and artificial neural networks (ANN) considering their application values.

Machine learning tools are widely used to support decision problems. The existing models predict the occurrence of construction disputes and provide decision-support information necessary to select the appropriate resolution strategy before a dispute occurs [21,22]. Other studies focus on investigating factors affecting the outcome of litigation, as well as on predicting the outcome of construction litigation itself [16,23–25]. In order to predict the optimal solution in a conflict situation, the authors applied various tools, including ANN [16,22,25] and DT [16,22,23], having based on data from a wide variety of sources: directly from courts, online databases, literature. The data was frequently collected from a wide variety of construction projects executed in many different countries and obtained from many different construction companies. Therefore, the novelty of the proposed method of a decision support is based on the historical dispute cases of only one contractor. What is more, predictions are based solely on time and financial data usually collected by a contractor.

The subject of the article is quantitative risk assessment in construction disputes based on machine learning tools. The article presents the most common causes of conflicts between parties of the construction contract, defines the background of the problem as well as introduces an example incorporating a real-life problem. By using DT and ANN the authors present application possibilities of the tools supporting the contractor's decision-making process in the conflict situation with a client.

The process of getting to the proposed decision support method is presented in Figure 1.

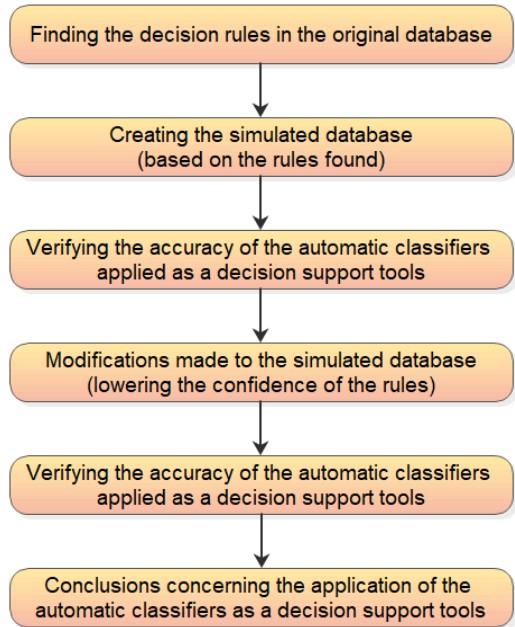

**Figure 1.** The process of getting to the proposed decision support method.

The applied tools, i.e., artificial neural networks (ANN), decision trees (DT), and association analysis are presented in Section 2. Then, the association rules concerning the provided real dataset on construction contracts problems are found. They are the base of a much wider database, simulated and described in Section 2.2. The full simulated database

is presented in Appendix A. Then, in Section 3, the accuracy of automatic classifiers is verified on that extended database. To model other, less structured cases the database is step by step modified, distorted and the accuracy of the classifiers is checked at every level of modifications. The results achieved in Section 3 are discussed in Section 4. There is also an example of application the proposed working-out the decision together with the proposed the risk read-out from the machine learning models that support the decision-making process. The findings are summarized and concluded in Section 5.

## 2. Materials and Methods

### 2.1. Supporting Tools

The main goal of the article is to find an optimal strategy for GC being in a conflict situation with a client, based on the historical data (real cases) regarding similar investments completed in the last six years. Considering the available data it was decided to use classification and regression trees based on their classification properties [26–29]. The second tool applied for calculations was artificial neural networks (multilayer perception MLP type). They were considered since the machine learning tool has been successfully applied in many construction problems supporting the optimal decision based on historical data and declared parameters [30–33].

#### 2.1.1. Decision Trees—Classifier

Classification and regression trees (C&RT) allow for both the creation of models to solve the regression problems (where the dependent variable is a quantitative feature) and solving classification problems (with qualitative dependent variable). The classic C&RT algorithm was popularized by Breiman et al. [28]. In the most general terms, the goal of analysis using the tree-building algorithm is to find a set of logical partitioning conditions, of type "if, then", leading to an unambiguous classification of objects [34].

There are three types of elements crating the decision tree model. The selected (by the built-in algorithm) attributes are split in the *decision nodes* (also called *split nodes* or *internal nodes*). The top split node can be named the *root node*. Each split creates two *branches* – the second type of the decision tree elements. At the end of each branch there is another split node or the *leaf node* (often called the *leaf* or the *end node*). The third type of the decision tree elements – leaf nodes – classify the target (dependent variable) [35]. A specific independent variable is assigned to each decision node together with its threshold value (the basis of a division on two branches). When the leaf node is reached, its content presents the expected value of dependent variable for independent variables meeting the rules found in split nodes [36]. The decision tree creates the flowchart that categorize the selected – by the built-in algorithm – types of data. The multi-end of the flowchart i.e. the leaves should contain the independent variable of one, predefined class. That is the aim of the algorithm [35].

The most critical parameter of the decision tree is its depth i.e. number of split nodes between the root node and a leaf. The deeper is the tree, the more accurately the output is classified. However, the risk of overfitting is higher then [26,37]. The decision trees are incapable of of predicting the continuous target. Nevertheless, this disadvantage can be overcome if the the range of the output values is limited and there are many leaves found [26,28].

Despite the above-mentioned limitations, the advantages of the decision tree method made it widely applied for classification and prediction problems, e.g., for

- rockburst prediction [38],
- predicting the compressive strength of cement-stabilized rammed earth [26],
- forecasting transport issues [36,39],
- energy demand modeling [35],
- monitoring the condition of rotating machinery [40],
- optimization of power systems [41],
- predicting the popularity of colleges [42] and many others.

Finding a decision tree for a specific set of data and pre-defined output does not require a high computational effort. The method can be applied for the both categorical and numerical types of data in one dataset. However, the greatest advantage of the decision tree method is clear structuring the input dataset – the independent variables. The subsets of input data supports appearance of a certain class (or value) of the output. The structure of the tree can be drawn in a form of a flowchart and can be easily interpreted by a user. So, the knowledge of machine learning issues is not critical to explain phenomena based on the decision trees found [35].

### 2.1.2. Artificial Neural Networks—Classifier

Artificial neural networks (ANNs) are a well-known branch of machine learning. The first attempts to apply ANN in construction took place in the early 1990s. Artificial neural networks were considered as a potential tool to support decision-making in civil engineering. They have been successfully applied in construction, supporting the optimal decision based on historical data and declared parameters.

In the area of construction, ANN were used, among others, to:

- forecasting the flow of costs in construction projects [30],
- estimating the construction costs of investment projects [31,33,43],
- support contractors' bidding decisions [44],
- evaluation of delays in the execution of construction contracts [45,46],
- calculating the cost of premises in multifamily housing, considering their various technical parameters [47],
- multicriteria optimization of the project of a residential building [48],
- predicting the effects of vibrations (e.g., damage or construction disasters) caused by road traffic [49],
- modeling of urban development, simulation of urban expansion, research into changes in the use of urban areas [50],
- increasing the efficiency of design and adaptation of municipal water infrastructure [51].

Feed-forward, multilayer neural networks are often used in publications focused on solving civil engineering problems. Historical data serves as training data: its analysis allows to identify the main factors characterizing and significantly impacting the given problem. Those factors are incorporated into the neural model as input variables. The training algorithm typically selected is one of the most popular ones [52]—the back-propagation, where the weights and biases are adjusted layer by layer from the output layer toward the input layer. The whole process is then repeated until a satisfactory error level is reached or becomes stationary. It was also applied in that case study.

### 2.1.3. Association Analysis—Rules Finding Tool

The association analysis, called also market basket analysis was originally invented to enhance the sales of supermarkets [53]. The contents of the clients' baskets—in the supermarkets—were searched to find the simultaneity of the appearance of specific goods. If found, it allows to modify prices or shelf layouts. The association analysis results are read through the two basic ratios: support (sup) and confidence (conf) defined below (1, 2) [54,55].

$$sup = \frac{n(B \rightarrow H)}{N}, \tag{1}$$

$$conf = \frac{n(B \rightarrow H)}{n(B)}, \tag{2}$$

where:

$B$   -   so-called body of the rule (the predecessor)
$H$   -   so-called head of the rule (the consequent)
$n(B{\rightarrow}H)$ number of cases of the simultaneous appearance of body and head

$N$ - total number of cases in the analyzed database

$n(B)$ - number of cases of the body appearance

The predecessor $B$ and the consequent $H$ are the states or phenomena. Their joint appearance is a subject of the association analysis. The rule, if $B$ appears, then $H$ also appears (denoted as $B \rightarrow H$) is described by the support and the confidence. The confidence of 100% (the highest possible) means that the appearance of $B$ makes, every time, $H$ also appear. To describe it as a strong rule, support of this rule has to be calculated. If the support equals to 1% (for the 100 rows database) it means that $B \rightarrow H$ happened only once. It can be by chance. There is no strict definition, the parameters of the rule can make it assessed as a strong one [56]. Every time it depends on the analyzed problem. The predecessor can be constructed as a conjunction of several conditions/states to be met (e.g., the temperature was rising a.m. and it reached 31 °C, and the atmospheric pressure was declining a.m.—it will be denoted $B = (b_1 \cap b_2 \cap b_3)$). Then the rule for appearance of an afternoon storm ($H$) can be calculated and assessed.

Nowadays, the market basket analysis applications cover a much wider area. Despite its original applications (they can be still found e.g., [57]) there is a spectrum of the association analysis applications. For example, it is applied in:

- social sciences for preferences searching [58],
- biology for variety of problems [59–61],
- meteorology for rainfall predictions [62],
- insurance for risk assessment [63].

The tool was also utilized for solving problems in civil engineering e.g., for:

- quality management in a precast concrete elements production [64],
- detection of bid-rigging in the construction industry [65],
- construction project risk assessment [46],
- traffic safety issues [66,67].

The rules that can be found in the database help to describe the analyzed processes or to find the critical elements of the processes. Just for those features, the association analysis is applied to analyze the original database and to simulate a much wider database analyzed then.

### 2.2. Databases

#### 2.2.1. Original Database and the Problem to Solve

Information presented in the article refers to real construction projects and was made available by the management of a large construction company operating in Europe. The company specializes mainly in "design-bid-build" and "design-build" project delivery systems as a substitute investor and a general contractor (GC), however, the examples below (Table 1) also include 'build' project delivery system. The form of settlement for each project is a lump sum.

The project X, which is the problem to be solved, is currently executed by a company in Poland. Due to significant defects in the project documentation provided by a client (IN), identified during works execution, it was necessary to extend the scope of the works (to a total value of 2.0 million monetary units). Having the detailed analysis of the consequences of completing the additional works, GC demanded from IN an annex to the contract, increasing the agreed lump-sum remuneration and extending the deadline. IN rejected GC's claims. At the stage of works execution, GC considered the following options: to stop further construction works or to continue works without the annex to the contract and guarantee of payment for the additional scope of work, i.e., pursuing the claim after works completion.

To identify the best course of action GC conducted research, using the methods of experts group, brainstorming and preliminary hazard analysis. The research included 12 experts—directly involved in the project (among them: the Project Manager, Site Man-

ager, Accounting Specialist, Contracts Specialist). Eleven of the experts 'preferred' to continue and one expert preferred to stop.

**Table 1.** Examples of historical data regarding the company's contracts in the conflict situation with GC and the problem to solve.

| Contract Number | Contract Value in Mln PLN | Planned Cost in Mln PLN | Planned Profit in Mln PLN | Financial Reserve in Mln PLN | Contract Scope | Planned Duration in Days | Direct Cost of Additional Works in Mln PLN | Delay in Days | Additional Fixed Cost for GC in Mln PLN/Day | Total Fixed Cost Increase in Mln PLN | Decision to Sue Client to Court? | Sentence |
|---|---|---|---|---|---|---|---|---|---|---|---|---|
| (i) | (v) | (c) | (p) | (r) | (k) | (t) | (a) | (d) | (u) | (q) | (s) | (w) |
| 1. | 22.50 | 20.00 | 1.500 | 1.00 | 0 | 180 | 0.530 | 35 | 0.02300 | 0.80500 | 1 | 1 |
| 2. | 6.30 | 5.100 | 0.800 | 0.400 | 0 | 135 | 0.100 | 48 | 0.00600 | 0.28800 | 0 | 0 |
| 3. | 44.80 | 43.000 | 1.100 | 0.700 | 0 | 320 | 0.760 | 52 | 0.00600 | 0.31200 | 1 | 1 |
| 4. | 1.27 | 1.020 | 0.200 | 0.050 | 1 | 25 | 0.015 | 5 | 0.00700 | 0.03500 | 0 | 0 |
| 5. | 32.26 | 28.940 | 2.420 | 0.900 | 0 | 332 | 2.200 | 98 | 0.01600 | 1.56800 | 1 | 1 |
| 6. | 18.60 | 16.650 | 1.250 | 0.700 | 0 | 195 | 0.790 | 32 | 0.02000 | 0.64000 | 1 | 1 |
| 7. | 5.53 | 4.984 | 0.305 | 0.243 | 1 | 93 | 0.065 | 10 | 0.00120 | 0.01200 | 0 | 0 |
| 8. | 7.25 | 6.690 | 0.410 | 0.150 | 1 | 68 | 0.094 | 51 | 0.00180 | 0.09180 | 1 | 0 |
| 9. | 4.62 | 4.131 | 0.370 | 0.120 | 1 | 82 | 0.042 | 14 | 0.00100 | 0.01400 | 0 | 0 |
| 10 | 2.75 | 2.540 | 0.130 | 0.080 | 0 | 60 | 0.073 | 3 | 0.00075 | 0.00225 | 0 | 0 |
| **Problem to Solve** | | | | | | | | | | | | |
| X. | 50.00 | 45.000 | 3.500 | 1.500 | 0 | 450 | 2.000 | 120 | 0.01000 | 1.2 | ? | ? |

*(r)*—the allowed cost increase; *(k)*—1 for 'Design and Build', 0 for 'Build' (based on design provided by a client); *(d)*—the recorded delay in days due to unplanned works; *(u)*—cost including contractual fine for breaching the contract deadline, fine for subcontractors for GC breaching the deadline (e.g., not making the site available for further works on time), the maintenance cost of the construction site, employees and equipment; *(q)*—the total fixed cost increase arisen from the delay $(d*u)$; *(s)*—1 for 'yes', 0 for 'no'; *(w)*—1 for 'the sentence favorable for a GC', 0 for 'the opposite cases'.

After an in-depth risk analysis of both solutions, GC decided to continue the investment. GC completed works in line with the signed contract, as well as the additional works what resulted in a higher cost and deadline extension. GC estimates the delay at 120 workdays. The previous course of negotiations indicated that IN would not sign the annex. This means that after the completion of the works IN will claim the contractual fine for GC's delay (most probably it will be claimed from GC's guarantee bond).

The decision that GC faces is whether after the completion of works to pursue its claims in court to enforce the fee for additional works, the contractual fines claimed by IN, as well as costs of the court case and lost benefits (benefits lost due to the extended time of works, e.g., GC could not start another contract timely, what has numerous consequences, including financial ones: GC had to pay a contractual fine to subcontractors and the investor—client). GC's previous practice indicates that in none of the analyzed cases (10 examples are presented in Table 1) at the stage of works execution had GC decided to go to court. Such decisions were made only after the project had been completed. At the same time, GC is aware that in case of litigation the claims may be rejected incurring a high additional cost.

### 2.2.2. The Rules Found

There is a substantial gap in the contracts' values (see Table 1). The contractor completed four contracts with a value over PLN 18.5 million and six contracts lower than PLN

7.3 mln. It can be observed, that none of the contracts of the value below PLN 7.25 million were taken into court. This can be written down as a Rule 1 (3).

$$(v_i < 7.25) \rightarrow (s_i = 0), \tag{3}$$

The confidence of the rule is the maximum one i.e., 100%. However, there are millions of similar rules, as the threshold of Rule 1 can be from the range PLN 6.30–7.25 million. For further calculation, the following form of the Rule 1 is considered.

$$(v_i \leq 6.30) \rightarrow (s_i = 0), \tag{4}$$

The opposite form of Rule 1 to its form presented as (5) is the following one:

$$(v_i > 6.30) \rightarrow (s_i = 1), \tag{5}$$

All these three rules presented in (3), (4), and (5) have 100% confidence as well as the rule presented below.

$$(v_i \geq 7.25) \rightarrow (s_i = 1), \tag{6}$$

Another finding is that taking the client to a court is successful if the contract value is PLN 7.25 million or higher. This Rule 2 can be written down as:

$$(v_i > 7.25) \rightarrow (w_i = 1), \tag{7}$$

As before, here the variations of this 100% confident Rule 2 are also possible, e.g.,

$$(v_i \geq 18.60) \rightarrow (w_i = 1), \tag{8}$$

This rule is true for this specific 10-row database. To get a favorable sentence, it is necessary to sue the client. Considering that Rule 3 can be formulated (see (9)).

$$((s_i = 1) \cap (v_i > 7.25)) \rightarrow (w_i = 1), \tag{9}$$

Although both rules (Rules 2 and 3) have 100% confidence, Rule 3 reflects the real conditions better. There are no contracts settled based on unit prices of the construction works. The idea of creating the reserve ($r$) is to use it to cover unpredicted costs arisen during the contract execution. It allows to reach the planned profit. It is hard to predict the value of additional works ($a$) and additional costs arisen from the delay in a contract completion date ($q$). However, it can be observed in Table 1 that these two values are greater than 0 in every contract. Based on that, it is checked if the reserve covers the sum of costs $a$ and $q$, by calculating the $e$ value for each contract:

$$e_i = r_i - (a_i + q_i), \tag{10}$$

The following Rule 4 is found:

$$(e_i < 0) \rightarrow (s_i = 1), \tag{11}$$

Every time the contractor had not reached the planned profit, they sued the client. The confidence of Rule 4 is 100%. Applying a favorable sentence as a consequence of the rule (it creates the Rule 5), the confidence then decreases to 80%. One time (out of five) the sentence was not favorable for the contractor. Rule 5—considering also the necessity of taking a client to a court before the sentence—can be presented as:

$$((s_i = 1) \cap (e_i < 0)) \rightarrow (w_i = 1), \tag{12}$$

There is one more, strong rule found: if a client is sued and the contract time extension is longer than 40% of the planned time, then the sentence is not favorable for the contractor.

This Rule 6 (presented in (12)) is called "a rule" but it is only one case in the database, and its confidence is 100%.

$$((s_i = 1) \cap (d_i/t_i \geq 0.4)) \rightarrow (w_i = 0), \tag{13}$$

There are certainly many other rules to discover even in such a small database with lower or much lower confidence, e.g.,

$$(k_i = 0) \rightarrow (w_i = 1), \tag{14}$$

The confidence of this Rule 7 is 67% (four contracts with favorable sentences out of 6 contracts based on designs provided by clients). Another meaningless rule (according to the authors' opinion) is:

$$(r_i \geq 0.7) \rightarrow (w_i = 1), \tag{15}$$

The confidence of the rule presented in (15) is 100%. Despite the theoretical strength of this rule, ensuring a favorable sentence by assuming high reserve before the contract is signed, is not reasonable. One of the rules with a contract scope as a consequence is Rule 8.

$$(v_i < 18.60) \rightarrow (k_i = 1), \tag{16}$$

This time the confidence of the rule is 67%. The basic statistics of the contractor's completed projects parameters are presented in Table 2.

**Table 2.** The basic statistics of the contractor's contract parameters.

|  | v | c | p | r | t | a | d | u | q |
|---|---|---|---|---|---|---|---|---|---|
| Min | 1.270 | 1.020 | 0.130 | 0.050 | 25.0 | 0.015 | 3.0 | 0.001 | 0.002 |
| Max | 44.800 | 43.000 | 2.420 | 1.000 | 332.0 | 2.200 | 98.0 | 0.023 | 1.568 |
| Mean | 14.588 | 13.306 | 0.849 | 0.434 | 149.0 | 0.467 | 34.8 | 0.008 | 0.377 |
| St. dev. | 14.626 | 13.798 | 0.728 | 0.360 | 107.2 | 0.681 | 29.2 | 0.008 | 0.504 |

The following rules were chosen for further analysis: Rule 1 (in form presented in (5)), Rule 3 (presented in (9)), Rule 4 (presented in (11)), Rule 5 (presented in (12)), Rule 6 (presented in (13)), the Rule 8 (presented in (16)). Their basic association analysis parameters, i.e., confidence, support, and lift are presented in Table 3. There, n serves as a number of contracts meeting the rule and m serves as a number of contracts meeting the condition of a rule's predecessor.

**Table 3.** The association analysis parameters for the selected rules.

| Rules | *n* | *m* | *conf* | *supp* | *lift* |
|---|---|---|---|---|---|
| Rule 1 | 5 | 5 | 100% | 50% | 2.0 |
| Rule 3 | 4 | 4 | 100% | 40% | 2.5 |
| Rule 4 | 5 | 5 | 100% | 50% | 2.0 |
| Rule 5 | 4 | 5 | 80% | 40% | 2.0 |
| Rule 6 | 1 | 1 | 100% | 10% | 10.0 |
| Rule 8 | 4 | 6 | 67% | 40% | 1.7 |

Although the selected rules are very strong, their meaning for the contractor having only 10 contracts completed in its portfolio is low. There are numerous companies with much higher experience. Willing to prove the usefulness of machine learning tools for working out the decision of taking a client to a court, it is necessary to create a much wider, simulated database. It was already proved in [21] and [22] that with the use of machine learning a high accuracy of disputes' results (or occurrence) can be achieved. However, these analyses refer to the disputes of many contractors being parties in them. The novelty of the prosed approach is making an automatic classification based on data collected by a

single contractor (gathered from their experience). It was decided to overcome the problem of confidentiality of this kind financial and other type of data, by simulating the database based on the rules found for a real 10-row database provided. The contractor—the database provider—avoids in this case disclosing a full set of managerial, financial information, that is crucial for their competitiveness. Certainly, publicly known the level of profit and the level of the reserve (for cost increase) will lower the competitiveness of any contractor. Therefore, the database was created, providing 100 rows of data. The base of the simulated database are the rules presented in Table 3.

### 2.2.3. The Simulated Database

The intermediate aim is to create a database that could simulate data concerning 100 completed construction contracts. The rules presented in Table 3 and the real contracts' basic statistics presented in Table 1 are the base. The simulated contract values $v^{(s)}$ are created based on the following Formula (17):

$$v_i^{(s)} = \min(v) + rnd \cdot (\max(v) - \min(v)), \tag{17}$$

where $rnd$ is a random value from the range (0; 1) based on the linear distribution. Similarly, $u^{(s)}$ values and $d^{(s)}$ values are calculated. To simulate the other contracts' parameters several ratios are calculated (based on real data) for each original contract. Their minimum and maximum values are presented in Table 4.

**Table 4.** The association analysis parameters for the selected rules.

| Extrema | c/v | p/v | r/p | a/v |
|---------|------|------|------|------|
| Minimum | 80.3% | 2.5% | 25.0% | 0.9% |
| Maximum | 96.0% | 15.7% | 79.7% | 6.8% |

Next, the following Formulas (18)–(21) are applied to calculate the simulated planned cost, planned profit, reserve, and cost of additional works.

$$c_i^{(s)} = v_i^{(s)} \cdot [\min(c/v) + rnd \cdot (\max(c/v) - \min(c/v))], \tag{18}$$

$$p_i^{(s)} = v_i^{(s)} \cdot [\min(p/v) + rnd \cdot (\max(p/v) - \min(p/v))], \tag{19}$$

$$r_i^{(s)} = p_i^{(s)} \cdot [\min(r/p) + rnd \cdot (\max(r/p) - \min(r/p))], \tag{20}$$

$$a_i^{(s)} = v_i^{(s)} \cdot [\min(a/v) + rnd \cdot (\max(a/v) - \min(a/v))], \tag{21}$$

To simulate the planned time the Pearson's correlation is calculated between the contract values and the planned time. As it was found that it equals 0.951 (quite high linear correlation), the simulated contract times are created through the following procedure:

- with the use of Microsoft Excel Solver the constant parameters aa and bb of a linear Equation (22) are found, for the real dataset (minimum absolute error was a target);
- the created linear function is used for calculating simulated contract times based on the simulated contracts' values;
- the calculated simulated contract times are manually modified to provide the Pearson's correlation 0.951 of the simulated times and the simulated contract's values.

$$t_i^{(s)} = aa \cdot v_i^{(s)} + bb, \tag{22}$$

Then, the time extension can be simulated. To do so, the histogram is presented of the ratio $d$ to $t$ for the original contracts (see Figure 2).

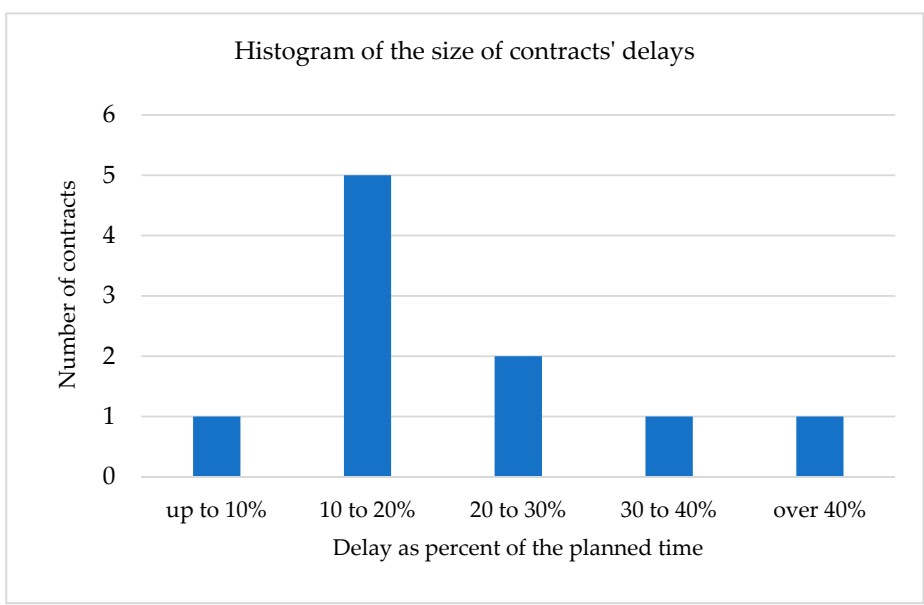

**Figure 2.** Histogram of contract delays calculated as percent of the planned time.

The sequence of simulating the values in a 100-row database is:

- simulating the contract values;
- simulating the planned cost, the planned profit, and the value of additional works (based on simulated contract values);
- simulating the reserve (based on simulated profit);
- simulating the unit cost;
- simulating the contracts' planned times (based on the procedure described above);
- simulating the extensions of time, based on simulated contracts' planned times, and keeping the shape of histogram presented in Figure 2;
- calculating the simulated total cost increase (d*u);
- assigning the simulated contracts' scopes according to Rule 8;
- assessing which clients of simulated contracts are sued (based on the intersection of cases arisen from Rule 1 and Rule 4; the rule is formulated as Rule 9 and presented in (23));
- assessing which simulated case won in a court (based on the intersection of cases arisen from Rule 3, Rule 5, and the rule opposite to Rule 6; the rule is formulated as Rule 10 and presented in (24)).

$$((v_i > 6.30) \cap (e_i < 0)) \rightarrow (s_i = 1), \tag{23}$$

$$((s_i = 1) \cap (v_i > 7.25) \cap (e_i < 0) \cap (d_i/t_i < 0.4)) \rightarrow (w_i = 1), \tag{24}$$

The simulated dataset is presented in Appendix A. The values of the selected parameters for the real 10-row database and the simulated 100-row database are presented in Table 5. The association analysis parameters of Rules 8, 9, and 10 for these two databases are compared in Table 6.

**Table 5.** Basic statistics of selected parameters calculated for the two databases.

| Database | Parameters | *c* | *e* | *c/v* | *p/v* | *r/p* | *a/v* |
|---|---|---|---|---|---|---|---|
| Real | Minimum | 1.270 | −2.868 | 0.803 | 0.025 | 0.250 | 0.009 |
| Simulated | Minimum | 2.389 | −3.175 | 0.804 | 0.032 | 0.256 | 0.008 |
| Real | Maximum | 44.800 | 0.166 | 0.960 | 0.157 | 0.797 | 0.068 |
| Simulated | Maximum | 44.605 | 3.062 | 0.956 | 0.157 | 0.790 | 0.068 |
| Real | Mean av. | 14.588 | −0.409 | 0.889 | 0.076 | 0.509 | 0.024 |
| Simulated | Mean av. | 23.123 | −0.092 | 0.879 | 0.096 | 0.518 | 0.038 |
| Real | Stand. Dev. | 14.626 | 0.905 | 0.049 | 0.039 | 0.176 | 0.018 |
| Simulated | Stand. Dev | 12.595 | 0.998 | 0.044 | 0.039 | 0.165 | 0.018 |

**Table 6.** The association analysis parameters for the selected rules, calculated for the databases.

| Database | Rule | *n* | *m* | *conf* | *supp* | *lift* | *N* [1] |
|---|---|---|---|---|---|---|---|
| Real | Rule 8 | 4 | 6 | 66.7% | 40.0% | 1.7 | 10 |
| Simulated | Rule 8 | 26 | 37 | 70.3% | 26.0% | 2.7 | 100 |
| Real | Rule 9 | 5 | 5 | 100.0% | 50.0% | 2.0 | 10 |
| Simulated | Rule 9 | 50 | 50 | 100.0% | 50.0% | 2.0 | 100 |
| Real | Rule 10 | 4 | 4 | 100.0% | 40.0% | 2.5 | 10 |
| Simulated | Rule 10 | 31 | 44 | 70.5% | 31.0% | 2.3 | 100 |

[1] N serves for the total number of cases in a database.

Based on the information presented in Tables 5 and 6, it can be stated that the patterns of these two databases are not identical. However, considering also almost identical Pearson's correlation coefficients (between contract values and planned times) the presented parameters are sufficiently close to assume that 100-row simulated database considers several, important dependencies in data discovered in the real, 10-row database. It is to emphasize that creating the simulated database perfectly reflecting 10-row database would be useless—the same relations will appear then in the original and the simulated databases. That is why the distributions of independent variables are not analyzed for original data and just a linear distribution is applied for simulation. The aim is achieved. As presented in Table 6, the databases are similar but not identical. Finally, the simulated database is large enough to apply machine learning tools and 100 contracts completed and it is still a real value for a construction company operating for several years.

## 3. Results

In the case of completing any contract by a certain company, its financial results are calculated. Finding them on the nonsatisfactory level, and having a real base to state that more works are executed than paid, the contractor's management board (or other entitled person or group of decision-makers) have to undertake the decision of taking a client to a court (if other methods are not successful) or not undertaking any action. There are several issues to be considered, but one of them is the history of the contractor's disputes. It can be done through an automatic model classifying the new case to the two subsets "the win" (w = 1) and "the loss" (w = 0) based on the past cases learned.

### 3.1. Classifications of the Simulated Data

The decision tree—presented in Figure 3—based on the following independent variables (*v, c, p, r, k, t, a, d, u, q, e*) 100% correctly classifies the cases. The algorithm built-in Statistica 13.1 (by Dell) software chose only *a, e, v, d,* and *k* variables to build the tree (for 10 cases in the leaf as a criterion of stop splitting the nodes).

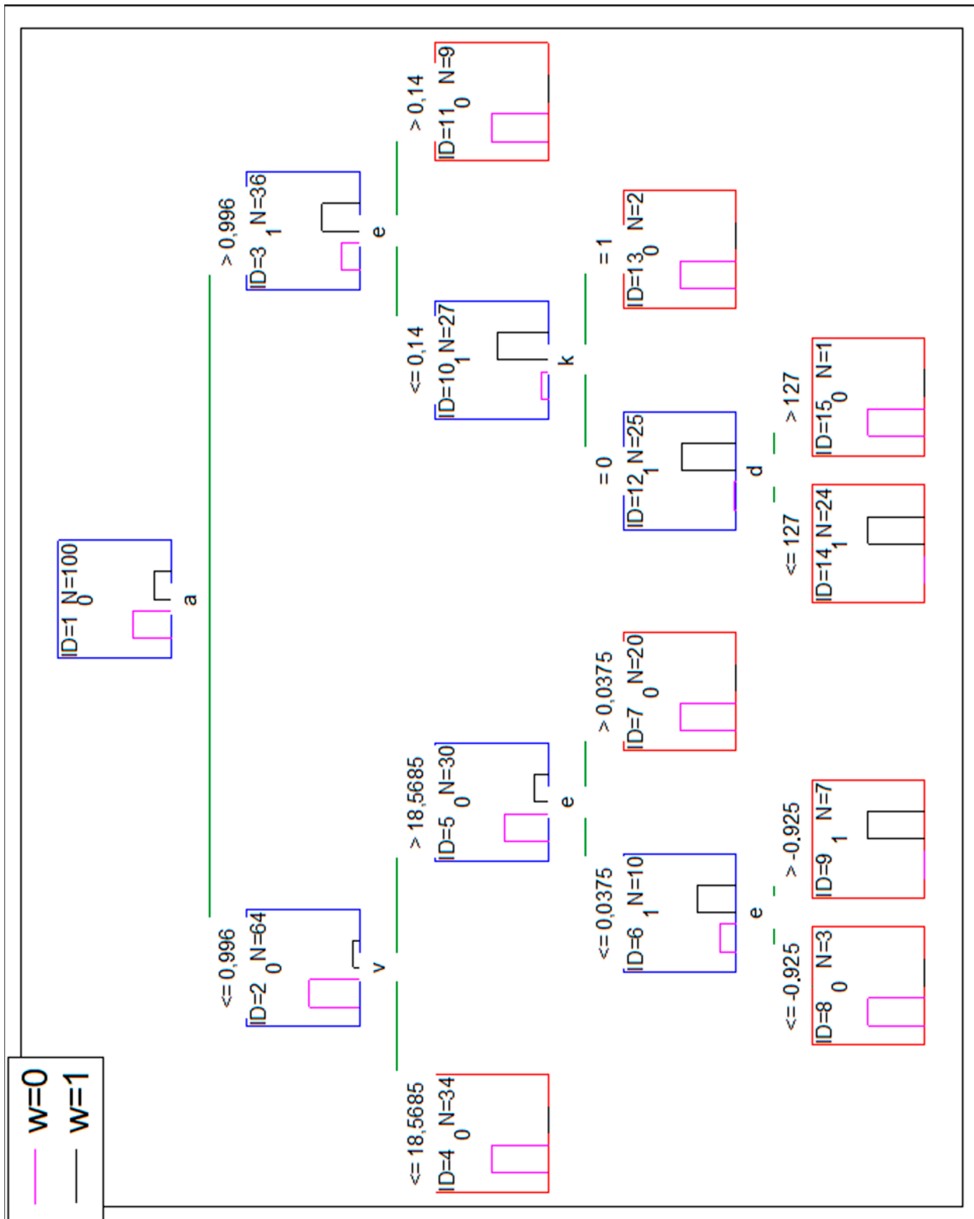

**Figure 3.** The decision tree applied to the simulated database.

A perfect classification is achieved, however, there are leaves with the very low number of cases (ID = 8–3 cases, ID = 13–2 cases, ID = 15–only one case). The reasoning based on them could be much more misleading than based on leaves with a higher number of cases (e.g., ID = 14 24 cases or ID = 9 7 cases there). It was decided to cut the tree by the condition of minimum five cases in a leaf. The result is presented in Figure 4.

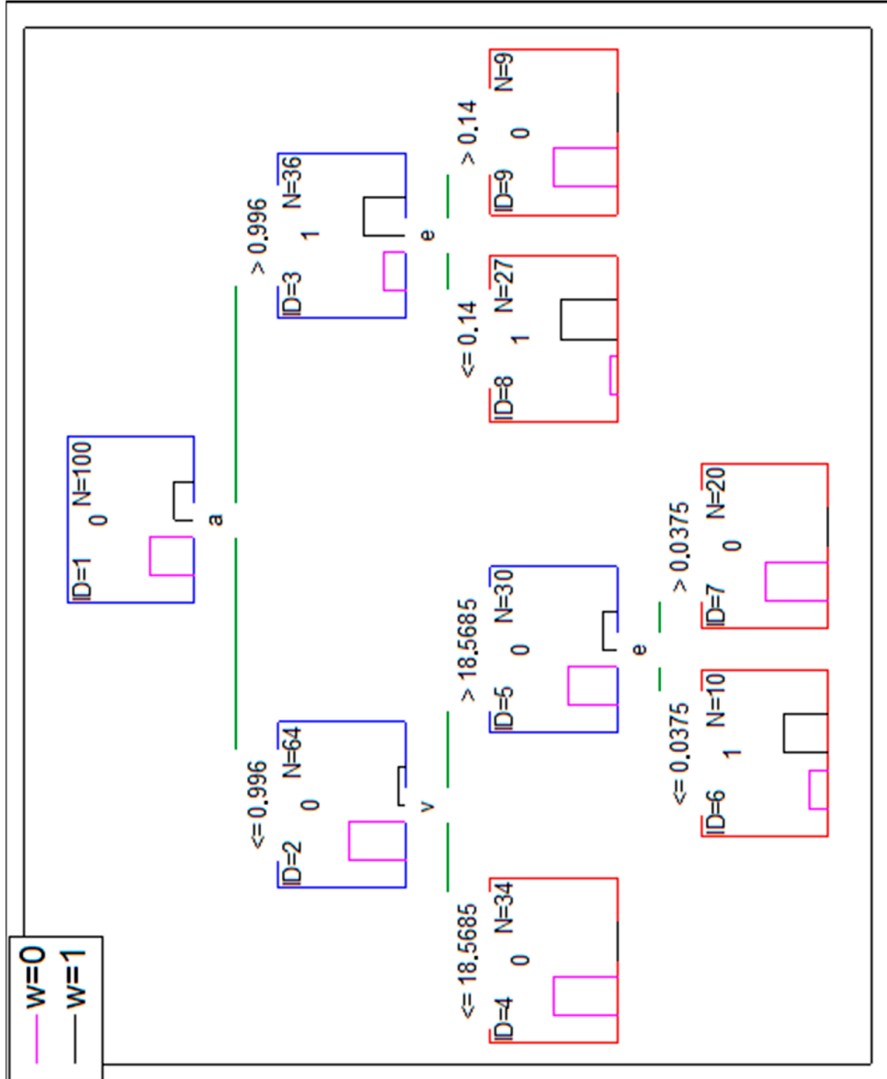

**Figure 4.** The decision tree applied to the simulated database with cut leaves.

This lowered the overall accuracy of DT to 94% (observed w = 0 six times is classified as w = 1). The confusion matrix is presented in Table 7.

**Table 7.** Confusion matrix for DT with cut leaves.

| Observed/Predicted | Predicted w = 1 | Predicted w = 0 |
|---|---|---|
| Observed *w = 1* | 31 | 0 |
| Observed *w = 0* | 6 | 63 |

Extending the number of inputs to DT by the parameters presented in Table 4 and *a + q* value does not increase the accuracy of DT classification.

Then, the artificial neural network is applied to the same database for this classification problem. Nonbinary data are normalized with the linear method. The software allows to search the best network (with only one hidden layer) through choosing a different number of neurons in a hidden layer, choosing the activation functions in the hidden and in the output layer and by choosing the training algorithm (while the weights of neurons are searched for minimizing the output error). There is the same set of inputs applied as for DT, however, there are 12 input neurons (ask is a category and has to be split on 0-1 or 1-0 pairs of input). According to only 100 rows in the database, the cross-validation process is applied (six folds are applied and every time five of the best classifying models are

saved). There are 31 cases in the database where w = 1. Therefore, the existence of five or six cases with w = 1 is provided (in the test subset and in the validating subset) for every fold. The test subset serves for finding the moment of stop training the network. The validating subset is applied for the assessment of the accuracy of the model. The overall accuracy 92.4% for the validating subset is achieved (92.9% for the "loss" category and 91.7% for the "won" category). The confusion matrix is presented in Table 8 (the results from the cross-validation are summed-up, not averaged).

**Table 8.** Confusion matrix for ANN.

| Observed/Predicted | Predicted w = 1 | Predicted w = 0 |
|---|---|---|
| Observed *w = 1* | 164 | 16 |
| Observed *w = 0* | 14 | 256 |

To improve the accuracy, the number of inputs is limited with different independent variables eliminated (one by one), but it does not provide higher accuracy of the ANN model. Moreover, considering only six independent variables taken from the ranking produced by the decision tree (presented in Figure 5) does not increase the accuracy of the ANN model either.

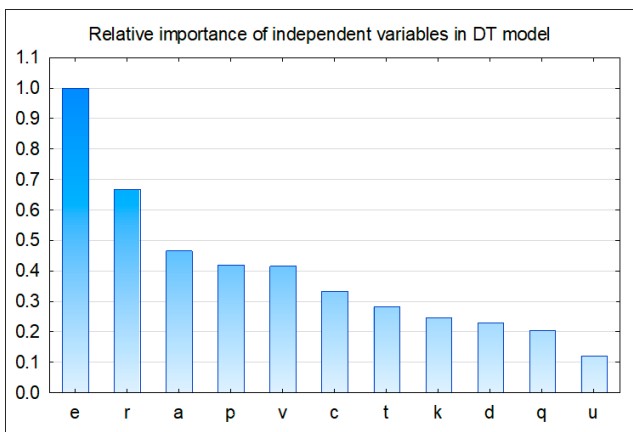

**Figure 5.** Relative importance of independent variables in DT model with cut leaves.

*3.2. Classifications of the Modified Simulated Data*

The proper classifications of the proposed tools is verified considering the mathematical point of view (e.g., by a cross-validation for ANN, by setting the conditions for splitting the nodes in DT). As the database is not real, it only reflects the reality at a certain level, it is decided to disturb the level of reflecting the reality by the simulated database i.e., the level of meeting the rules found in the original database. To verify the classification properties of DT and ANN for less structured data, the original simulated database is modified (this modification is further noted as mod-1). Approximately 20% of cases with w = 1 (six of them) are changed by assigning to them w = 0 (in rows 23, 42, 52, 57, 85, 96). Simultaneously, six cases with s = 1 and w = 0 are changed by assigning w = 1 to them (in rows 1, 7, 11, 41, 54, 56). The mod-1 lowers the accuracy of a DT. Based on the same set of independent variables (and the same criteria of the stop), the overall accuracy of DT classification is 92% (eight cases are misclassified). The confusion matrix is presented in Table 9.

**Table 9.** Confusion matrix for DT for mod-1.

| Observed/Predicted | Predicted w = 1 | Predicted w = 0 |
|---|---|---|
| Observed *w = 1* | 30 | 1 |
| Observed *w = 0* | 7 | 62 |

The structure of the tree is presented in Figure 6.

**Figure 6.** The decision tree applied to mod-1 database.

Searching for an increase in DT accuracy the types of inputs are extended considering the ratios presented in Table 4, as well as, an additional type of input is created equal to *a+q* (additional cost arisen from a contract delay and the cost of additional works are added). Then the confusion matrix is as presented in Table 10. The decision tree with its structure is presented in Appendix B as Figure A1.

**Table 10.** Confusion matrix for DT (with extended input) for mod-1.

| Observed/Predicted | Predicted w = 1 | Predicted w = 0 |
|---|---|---|
| Observed *w = 1* | 31 | 0 |
| Observed *w = 0* | 9 | 60 |

Similarly to the previous attempt to the classification, the accuracy of ANN classifications is searched. The confusion matrix from six folds (and five of the best classifying models for each fold) is presented in Table 11.

**Table 11.** Confusion matrix for ANN, for mod-1.

| Observed/Predicted | Predicted w = 1 | Predicted w = 0 |
|---|---|---|
| Observed *w = 1* | 125 | 40 |
| Observed *w = 0* | 15 | 285 |

Additionally, this time the attempts of reducing the number of inputs do not lower classification errors. Then, the database created—named as mod-1—is modified once more. The next six cases with w = 1 are changed by assigning w = 0 (in rows 3, 32, 46, 58 68, 82) and the other six cases with s = 1 and w = 0 are changed to w = 1 (in rows 4, 18, 19, 30, 44, 66). Therefore, next, near 20% of cases are modified for the new database (named mod-2) to differ more from the original one—created with the application of the rules. The DT found for these data is presented in Figure A2 (see Appendix B), but the confusion matrix as Table 12 is presented below. The results from DT with extended data (presented in Figure A3) produce exactly the same confusion matrix as presented in Table 12.

**Table 12.** Confusion matrix for DT for mod-2.

| Observed/Predicted | Predicted w = 1 | Predicted w = 0 |
|---|---|---|
| Observed *w = 1* | 29 | 2 |
| Observed *w = 0* | 5 | 64 |

The confusion matrix—results from ANN classifications—is presented in Table 13.

**Table 13.** Confusion matrix for ANN, for mod-2.

| Observed/Predicted | Predicted w = 1 | Predicted w = 0 |
|---|---|---|
| Observed *w = 1* | 96 | 39 |
| Observed *w = 0* | 17 | 298 |

The third modification made the database (named mod-3) different from the original one in approximately 60% of cases where w = 1. This time the next six cases (rows 12, 20, 38, 75, 92, 95) are changed (w = 1 to w = 0) and oppositely for the cases where s = 1 (rows 28, 55, 64, 69, 77, 88) w = 0 is changed to w = 1. The DT found is presented in Figure A4. The confusion matrix is presented in Table 14 below. Table 15 contains the confusion matrix based on the decision tree found for the extended input. This DT is presented in Figure A5.

**Table 14.** Confusion matrix for DT for mod-3.

| Observed/Predicted | Predicted w = 1 | Predicted w = 0 |
|---|---|---|
| Observed *w = 1* | 30 | 1 |
| Observed *w = 0* | 8 | 61 |

**Table 15.** Confusion matrix for DT (with extended input) for mod-3.

| Observed/Predicted | Predicted w = 1 | Predicted w = 0 |
|---|---|---|
| Observed *w = 1* | 30 | 1 |
| Observed *w = 0* | 7 | 62 |

The confusion matrix—results from ANN classifications—is presented in Table 16.

**Table 16.** Confusion matrix for ANN, for mod-3.

| Observed/Predicted | Predicted w = 1 | Predicted w = 0 |
|---|---|---|
| Observed $w = 1$ | 44 | 76 |
| Observed $w = 0$ | 21 | 309 |

## 4. Discussion

To interpret the results obtained by the tools applied, it is necessary to analyze the confusion matrix and the corresponding indicators to assess the diagnostic value of the classification.

In the field of machine learning and specifically in the problem of statistical classification, a confusion matrix is a table layout allowing visualization of the performance of an algorithm [68]. In this case, each row of the matrix represents the examples in an actual (observed) class, while each column represents the examples in the predicted class. The four possible outcomes of the matrix are:

- True Positive (TP),
- True Negative (TN),
- False Positive (FP; type I error or underestimation),
- False Negative (FN; type II error or overestimation).

Considering the problem analyzed, which is the conflict between the contractor and the client, the confusion matrix presented in Table 17 indicates the possible variants of strategy from the contractor's perspective.

**Table 17.** Confusion matrix—the possible variants of strategy from the contractor's perspective (interpretation of results).

| Observed/Predicted | Predicted w = 1 | Predicted w = 0 |
|---|---|---|
| Observed $w = 1$ | **TP**—decision **to sue** a client to the court confirmed by the model | **FN**—decision **not to sue** a client; potentially lost benefits from winning the cause |
| Observed $w = 0$ | **FP**—decision **to sue** a client; potentially lost cause; unnecessary legal costs and the perception of the company as an antagonist | **TN**—decision **not to sue** a client to the court confirmed by the model |

The effectiveness of the classification performed through DTs and ANNs was evaluated in terms of accuracy, recall and specificity (Formulas (25)–(27)) [69].

$$Accuracy = \frac{TP + TN}{TP + FP + TN + FN}, \tag{25}$$

$$Recall = \frac{TP}{TP + FN}, \tag{26}$$

$$Specificity = \frac{TN}{FP + TN}, \tag{27}$$

Accuracy (ACC) indicates the proportion of correct classifications, however, it may yield misleading results if the data set is unbalanced [68]. Hence, as a complement, it is advisable to analyze recall, which is a true positive rate (probability of detection), as well as specificity, which is a true negative rate. All the results presented in Section 3 are transformed to the following ratios: ACC, recall and specificity. They are presented in Table 18.

**Table 18.** Classification performance of ANN and DT for different datasets.

| Database | Classifier | Accuracy | Recall | Specificity |
|---|---|---|---|---|
| mod-0 [1] | ANN | 0.933 | 0.911 | 0.948 |
| mod-0 | DT | 0.940 | 1.000 | 0.913 |
| mod-1 | ANN | 0.882 | 0.758 | 0.950 |
| mod-1 | DT | 0.920 | 0.968 | 0.899 |
| mod-2 | ANN | 0.876 | 0.711 | 0.946 |
| mod-2 | DT (ext. input)) | 0.930 | 0.935 | 0.928 |
| mod-3 | ANN | 0.784 | 0.367 | 0.936 |
| mod-3 | DT | 0.920 | 0.968 | 0.899 |

[1] mod-0 is used for the simulated, but not modified dataset.

For every type of dataset, the DT tool outperforms the ANN classifier when accuracy and recall are considered. The specificity is better for the ANN classifier, for each dataset. The results presented separately for ANN and DT (see Figures 7 and 8).

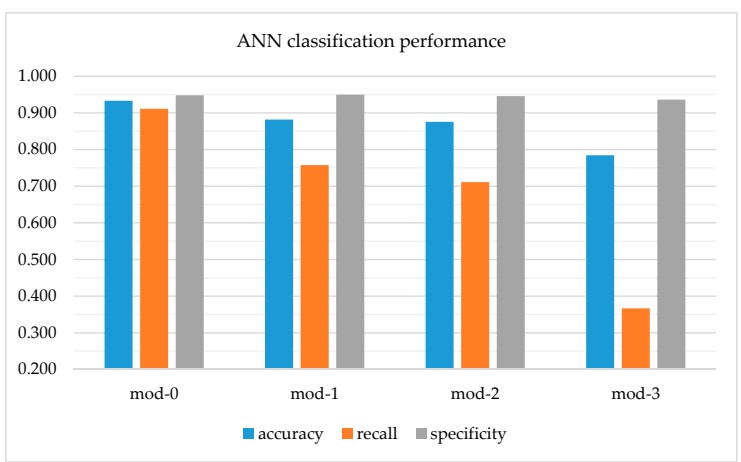

**Figure 7.** Performance of ANN classifier.

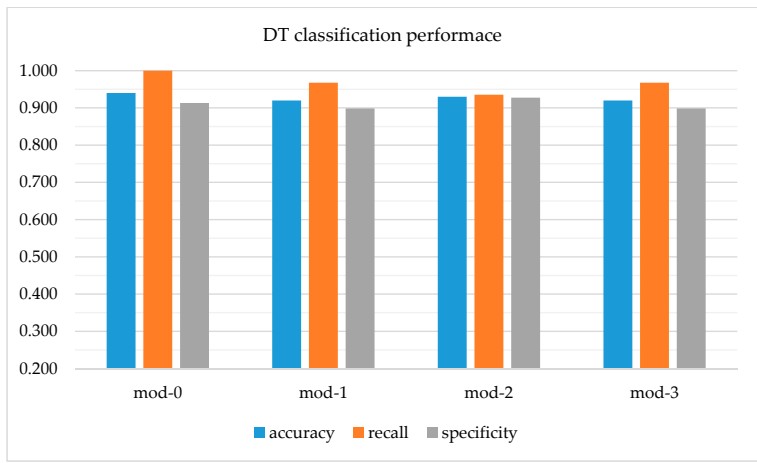

**Figure 8.** Performance of DT classifier.

The level of distortion of the data from the simulated dataset (mod-0) based on the rules found in the original dataset increases from mod 1 to mod-3. It influences a lot the recall of ANN. It decreases rapidly, while the decrease of the specificity is not considerable. It is due to the distortions made to the cases with s = 1 (clients sued). The subset of 69 cases with s = 0 is not modified for the purpose of creating mod-1 to mod-3. This made the specificity of ANN high. The recall of decision trees presents much higher resistance for

the distortions made to the datasets. Their classifications (when w = 1 is presented as the result) are not perfect (as for mod-0). Nevertheless, the levels of recall are high, every time above 93.5% for mod-1 and mod-3 (and 100% for mod-0). Recall and specificity for DT and ANN are compared in Figure 9.

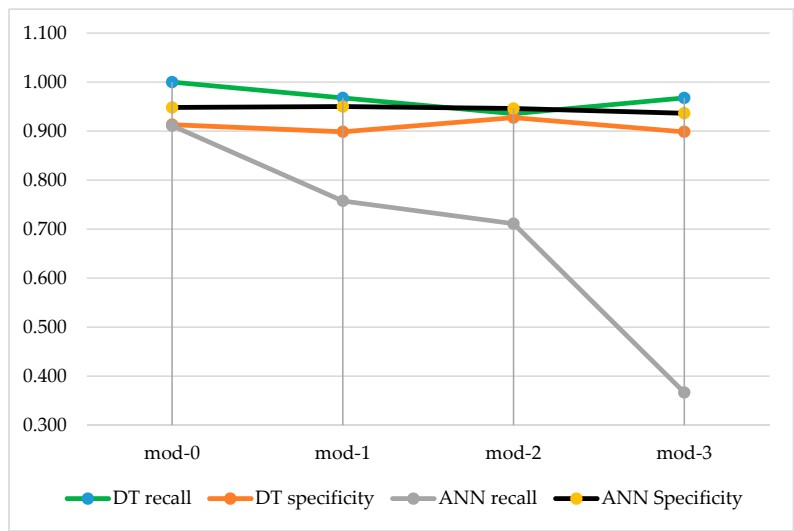

**Figure 9.** The comparison of recall and specificity for DT and ANN.

We analyze the risk of failure i.e., wrong decision based on ANN classification, assuming that the mod-1 database reflects the real case. If the classifier predicts w = 1, it is suggested to make the decision of taking a client to a court. Its recall (presented in Table 18) is 75.8%. The false-positive rate (FPR) [68,70] is then:

$$FPR = \frac{FP}{FP + TN} = 1 - recall, \tag{28}$$

and it is equal 24.2%. Then, there is a danger of losing the case in a court. Therefore, it is a risk of wrong decision undertaken on the basis of the supporting model (w = 1 is predicted). Similarly, when w = 0 is suggested by the decision supporting tool, the risk of wrong decision (based on ANN classification result) is equal to false-negative rate (FNR) defined as:

$$FNR = \frac{FN}{FN + TP} = 1 - specificity, \tag{29}$$

As the specificity of ANN for mod-1 is 95.0%, the risk of making the wrong decision is 5.0%. In case of materializing such risk (and not taking a client to a court), GC would lose benefits from a potential win in a court.

A similar reasoning can be made with a DT use, but it is recommended to utilize the feature of DT for a clear presentation of the process of classification in a form of a tree. Based on the same assumption (mod-1 represents a real case), the parameters of an analyzed project should be matched with the set of conditions of the tree (presented in Figure 6) until a leaf is reached. If the reached leaf suggests w = 1, e.g., it is ID = 12 leaf, the risk of a wrong decision is 50%, but for leaf ID = 12 the risk is 6/19 = 31.6%. However, it would be a risk-free decision if based on ID = 9 leaf.

It is recommended that the decision of taking a client to a court is supported by both models (ANN and DT). In case of agreement between suggested decisions from both models, the suggestions may be considered. The opposite suggestions found with these two models require comparison of the risk of each suggestion (calculated in different ways for ANN and DT), as well as, taking into account the policy of a specific GC. In case of w = 0 falsely suggested, GC will lose potential benefits. In case of falsely suggested w = 1, additional, not covered cost will be engaged without any benefit.

Analyzing the problem X (presented in Table 1 and described in Section 2.2.1.) it can be found that (considering the original 10-row dataset) the project X data meet at least two rules found. The project X value $v_x$ is greater than 18.60, so based on the rule found and presented in (8) there is 100% confidence of a favorable sentence in a court. Additionally, the rule presented in (12) is met. The total additional costs of the project X are not covered by the financial reserve. This makes the confidence 100% of winning the case in a court. However, there is the rule (13) which is not met by the parameters of the project X i.e., the delay of the completion date is lower than 40% of the scheduled time. Therefore, this rule indicating the loss in a court (if met), is not met. It is one more argument to sue a client.

Let us then assume that the project X was executed by a company, for which the mod-0 dataset is valid. If DT (presented in Figure A1) is used, the following way to reach a leaf should be taken, starting from split node ID = 1: $e_x < 0.0005$, the node ID = 2 should be considered then; $t_x > 127.5$, the node ID = 5 should be considered then; $d_x < 147$, the node ID = 8 should be considered then; $p_x > 2.95$. The leaf ID = 11 is reached. The read-out from the leaf is as follows: there are 14 projects meeting the same criteria (stated in split nodes), and for all of those disputes there were favorable court sentences. DT suggests the risk-free decision of taking a client to a court in case of project X. ANN classification confirms w = 1 for the project X input with recall 75.8%, so there is a 24.2% risk of a wrong decision (if the suggestion is considered).

## 5. Conclusions

The full set of data could not be provided by the construction company for the protection of the source of their competitive advantage. It is important to remember that the findings and conclusions are based on the simulated database reflecting the dependencies found for the original, 10-row database. However, as a quite high level of reflecting the reality is kept, it can be concluded that the findings underlined below can be confirmed in the real case of any other construction company (based on a full set of usually confidential managerial type data). The novelty of the proposed method of a decision support is based on the historical dispute cases of one contractor. Secondly, predictions of the effect of a dispute with a client are based solely on time and financial data usually collected by a contractor. The risk level of a particular decision is assessed as the probability of misclassifying the result of a dispute by ANN or DT classifier. The accuracy of decision trees and artificial neural networks is over 93%. The main findings based on DT and ANN application can be bulleted:

- the accuracy is not a sufficient measure for the comparison of DT and ANN performance,
- the risk of a wrong decision based on ANN can be measured by false-positive rate or false-negative rate (dependently on predicted class),
- the higher the level of data distortion, the lower the recall of ANN,
- even the recall of DT is quite stable (if data are distorted), it reflects an average performance; while reasoning for a particular new case, the risk of a wrong decision is calculated based on the leaf (so it can vary from case, to case),
- working out the predictions from both tools allows for a more precise assessment of the risk of the decision (based on the consistency of the predictions), as the ANN's error of classification is an average one and the DT's error of classification may vary, depending on a leaf (relevant to the case).

The property of the method to find real relations (input–output) is confirmed by presented lowering accuracy of ANN for the databases with increasing level of distortion (from structured data, reflecting the original ones). The other group of advantages of the worked-out method is related to its ease of application:

- predicting the outcome of a construction dispute can be successfully introduced practically in any type of a company if they have sufficient historical record of their disputes (however, it is confirmed for a construction company),
- the method is based on time and financial type of data that are usually recorded—the historical cases can be easily retrieved,
- the reliability of such tool increases with the size of the input database,

- the method can be applied for the projects (and disputes) executed in one state (for the contractor's contracts executed there).

There is a significant part of information that each contractor does not share, as their competitive advantages depend on that. There can be many more financial types of information (not shared for this article) that influence the result. They should be used in real applications, as algorithms built in decision trees will choose the most influential independent variables. The method may bring weaker results if the completed projects are completed in different states with different legal systems and different construction practices. It is recommended to avoid such a varied data used as input.

The concept of quantifying the risk with the use of machine learning tools and the probability of misclassifications calculated through them will be explored by the authors.

**Author Contributions:** Conceptualization, B.G., H.A. and M.A.; methodology, H.A. and M.A.; software, H.A.; validation, H.A.; formal analysis, B.G., H.A. and M.A.; investigation, B.G.; resources, B.G., H.A. and M.A.; data curation, B.G. and H.A.; writing—original draft preparation, B.G., H.A. and M.A.; writing—review and editing, M.A. and H.A..; visualization, H.A.; supervision, B.G., H.A. and M.A.; project administration, M.A. All authors have read and agreed to the published version of the manuscript.

**Funding:** This research received no external funding.

**Institutional Review Board Statement:** Not applicable.

**Informed Consent Statement:** Not applicable.

**Data Availability Statement:** All data supporting reported results can be found in Appendix A.

**Conflicts of Interest:** The authors declare no conflict of interest.

## Appendix A

**Table A1.** The simulated dataset based on rules found for the original 10-row dataset.

| i | v | c | p | r | k | t | a | d | u | q | s | w | e | c/v | p/v | r/p | a/v | a+q |
|---|---|---|---|---|---|---|---|---|---|---|---|---|---|---|---|---|---|---|
| 1 | 17.696 | 15.810 | 1.286 | 0.898 | 1 | 171 | 1.115 | 26 | 0.01929 | 0.502 | 1 | 0 | −0.719 | 0.893 | 0.073 | 0.698 | 0.063 | 1.617 |
| 2 | 26.721 | 24.703 | 3.746 | 1.255 | 0 | 180 | 0.946 | 44 | 0.00459 | 0.202 | 0 | 0 | 0.108 | 0.925 | 0.140 | 0.335 | 0.035 | 1.148 |
| 3 | 34.397 | 28.091 | 1.729 | 0.538 | 0 | 310 | 1.723 | 21 | 0.00488 | 0.102 | 1 | 1 | −1.287 | 0.817 | 0.050 | 0.311 | 0.050 | 1.825 |
| 4 | 23.937 | 22.789 | 1.996 | 0.974 | 0 | 170 | 0.893 | 107 | 0.00977 | 1.045 | 1 | 0 | −0.964 | 0.952 | 0.083 | 0.488 | 0.037 | 1.938 |
| 5 | 24.560 | 23.086 | 3.154 | 1.329 | 0 | 227 | 0.910 | 34 | 0.01298 | 0.441 | 1 | 1 | −0.023 | 0.940 | 0.128 | 0.421 | 0.037 | 1.351 |
| 6 | 42.589 | 37.697 | 6.071 | 3.401 | 0 | 290 | 1.783 | 29 | 0.00925 | 0.268 | 0 | 0 | 1.350 | 0.885 | 0.143 | 0.560 | 0.042 | 2.051 |
| 7 | 23.968 | 22.012 | 1.390 | 0.663 | 0 | 212 | 0.611 | 126 | 0.01443 | 1.818 | 1 | 0 | −1.766 | 0.918 | 0.058 | 0.477 | 0.025 | 2.429 |
| 8 | 3.550 | 3.253 | 0.358 | 0.228 | 1 | 51 | 0.122 | 5 | 0.00292 | 0.015 | 0 | 0 | 0.091 | 0.916 | 0.101 | 0.636 | 0.034 | 0.137 |
| 9 | 3.003 | 2.426 | 0.216 | 0.058 | 1 | 42 | 0.094 | 4 | 0.00791 | 0.032 | 0 | 0 | −0.068 | 0.808 | 0.072 | 0.266 | 0.031 | 0.126 |
| 10 | 39.738 | 35.373 | 4.860 | 3.088 | 0 | 260 | 0.618 | 27 | 0.00713 | 0.193 | 0 | 0 | 2.277 | 0.890 | 0.122 | 0.635 | 0.016 | 0.811 |
| 11 | 12.070 | 11.019 | 1.493 | 0.498 | 1 | 129 | 0.737 | 23 | 0.00116 | 0.027 | 1 | 0 | −0.265 | 0.913 | 0.124 | 0.334 | 0.061 | 0.764 |
| 12 | 38.744 | 35.412 | 3.585 | 1.568 | 0 | 330 | 2.400 | 82 | 0.01048 | 0.859 | 1 | 1 | −1.692 | 0.914 | 0.093 | 0.437 | 0.062 | 3.259 |
| 13 | 4.031 | 3.745 | 0.288 | 0.227 | 0 | 58 | 0.256 | 9 | 0.01826 | 0.164 | 0 | 0 | −0.193 | 0.929 | 0.071 | 0.790 | 0.064 | 0.420 |
| 14 | 4.153 | 3.844 | 0.453 | 0.133 | 0 | 60 | 0.148 | 28 | 0.01817 | 0.509 | 0 | 0 | −0.524 | 0.926 | 0.109 | 0.293 | 0.036 | 0.657 |
| 15 | 38.406 | 34.724 | 6.026 | 3.306 | 0 | 261 | 2.398 | 64 | 0.00529 | 0.338 | 0 | 0 | 0.570 | 0.904 | 0.157 | 0.549 | 0.062 | 2.736 |
| 16 | 33.276 | 31.136 | 2.704 | 1.593 | 0 | 307 | 0.700 | 52 | 0.00771 | 0.401 | 0 | 0 | 0.493 | 0.936 | 0.081 | 0.589 | 0.021 | 1.101 |
| 17 | 30.608 | 26.351 | 3.718 | 2.150 | 0 | 272 | 0.313 | 37 | 0.01455 | 0.538 | 0 | 0 | 1.299 | 0.861 | 0.121 | 0.578 | 0.010 | 0.851 |
| 18 | 13.220 | 11.198 | 0.465 | 0.130 | 1 | 90 | 0.605 | 10 | 0.00733 | 0.073 | 1 | 0 | −0.548 | 0.847 | 0.035 | 0.280 | 0.046 | 0.678 |
| 19 | 16.061 | 14.742 | 1.001 | 0.754 | 0 | 114 | 0.302 | 59 | 0.01093 | 0.645 | 1 | 0 | −0.193 | 0.918 | 0.062 | 0.753 | 0.019 | 0.947 |
| 20 | 35.880 | 33.486 | 2.785 | 0.932 | 0 | 327 | 2.359 | 36 | 0.01163 | 0.419 | 1 | 1 | −1.846 | 0.933 | 0.078 | 0.335 | 0.066 | 2.778 |
| 21 | 23.656 | 22.385 | 3.139 | 1.905 | 0 | 168 | 0.310 | 41 | 0.01105 | 0.453 | 0 | 0 | 1.142 | 0.946 | 0.133 | 0.607 | 0.013 | 0.763 |
| 22 | 41.862 | 33.909 | 6.411 | 4.859 | 0 | 290 | 0.945 | 48 | 0.01774 | 0.851 | 0 | 0 | 3.062 | 0.810 | 0.153 | 0.758 | 0.023 | 1.796 |
| 23 | 31.030 | 28.135 | 1.344 | 0.682 | 0 | 293 | 1.618 | 39 | 0.00813 | 0.317 | 1 | 1 | −1.254 | 0.907 | 0.043 | 0.507 | 0.052 | 1.935 |
| 24 | 2.584 | 2.425 | 0.311 | 0.198 | 1 | 38 | 0.050 | 11 | 0.00293 | 0.032 | 0 | 0 | 0.115 | 0.939 | 0.120 | 0.635 | 0.019 | 0.082 |
| 25 | 34.920 | 29.529 | 1.427 | 0.861 | 0 | 242 | 0.719 | 12 | 0.00211 | 0.025 | 0 | 0 | 0.117 | 0.846 | 0.041 | 0.603 | 0.021 | 0.744 |
| 26 | 6.293 | 5.472 | 0.608 | 0.331 | 1 | 75 | 0.161 | 40 | 0.01588 | 0.635 | 0 | 0 | −0.465 | 0.869 | 0.097 | 0.545 | 0.026 | 0.796 |
| 27 | 19.665 | 17.550 | 1.776 | 1.294 | 0 | 130 | 0.808 | 18 | 0.00325 | 0.059 | 0 | 0 | 0.427 | 0.892 | 0.090 | 0.729 | 0.041 | 0.867 |

**Table A1.** *Cont.*

| i | v | c | p | r | k | t | a | d | u | q | s | w | e | c/v | p/v | r/p | a/v | a+q |
|---|---|---|---|---|---|---|---|---|---|---|---|---|---|---|---|---|---|---|
| 28 | 29.120 | 23.644 | 3.829 | 1.230 | 0 | 265 | 0.809 | 168 | 0.02141 | 3.596 | 1 | 0 | −3.175 | 0.812 | 0.131 | 0.321 | 0.028 | 4.405 |
| 29 | 21.042 | 17.876 | 1.344 | 0.564 | 0 | 202 | 0.257 | 62 | 0.01781 | 1.104 | 1 | 1 | −0.797 | 0.850 | 0.064 | 0.419 | 0.012 | 1.361 |
| 30 | 39.507 | 33.604 | 3.196 | 2.089 | 0 | 332 | 1.246 | 171 | 0.00612 | 1.047 | 1 | 0 | −0.203 | 0.851 | 0.081 | 0.654 | 0.032 | 2.293 |
| 31 | 31.380 | 26.680 | 4.160 | 1.274 | 0 | 208 | 0.952 | 40 | 0.02064 | 0.825 | 1 | 1 | −0.503 | 0.850 | 0.133 | 0.306 | 0.030 | 1.777 |
| 32 | 23.981 | 20.655 | 3.247 | 1.931 | 0 | 215 | 0.997 | 50 | 0.01868 | 0.934 | 1 | 1 | −0.001 | 0.861 | 0.135 | 0.595 | 0.042 | 1.931 |
| 33 | 16.781 | 13.987 | 1.776 | 0.941 | 0 | 120 | 0.458 | 29 | 0.00762 | 0.221 | 0 | 0 | 0.262 | 0.833 | 0.106 | 0.530 | 0.027 | 0.679 |
| 34 | 28.938 | 27.573 | 2.991 | 1.020 | 0 | 261 | 1.788 | 40 | 0.00594 | 0.238 | 1 | 1 | −1.005 | 0.953 | 0.103 | 0.341 | 0.062 | 2.026 |
| 35 | 17.205 | 15.738 | 1.774 | 1.365 | 1 | 121 | 1.052 | 14 | 0.00227 | 0.032 | 0 | 0 | 0.281 | 0.915 | 0.103 | 0.769 | 0.061 | 1.084 |
| 36 | 26.369 | 24.329 | 3.662 | 2.191 | 0 | 236 | 1.311 | 56 | 0.00835 | 0.468 | 0 | 0 | 0.412 | 0.923 | 0.139 | 0.598 | 0.050 | 1.779 |
| 37 | 29.450 | 24.652 | 3.734 | 2.537 | 0 | 201 | 1.828 | 50 | 0.02294 | 1.147 | 1 | 1 | −0.438 | 0.837 | 0.127 | 0.679 | 0.062 | 2.975 |
| 38 | 19.702 | 16.648 | 0.931 | 0.553 | 0 | 190 | 1.150 | 43 | 0.01866 | 0.802 | 1 | 1 | −1.399 | 0.845 | 0.047 | 0.594 | 0.058 | 1.952 |
| 39 | 4.751 | 4.301 | 0.737 | 0.307 | 1 | 45 | 0.302 | 8 | 0.01594 | 0.127 | 0 | 0 | −0.122 | 0.905 | 0.155 | 0.417 | 0.064 | 0.429 |
| 40 | 37.921 | 30.893 | 1.991 | 1.383 | 0 | 269 | 1.242 | 83 | 0.01066 | 0.885 | 1 | 1 | −0.744 | 0.815 | 0.053 | 0.695 | 0.033 | 2.127 |
| 41 | 12.868 | 10.431 | 0.891 | 0.295 | 1 | 90 | 0.386 | 17 | 0.00492 | 0.084 | 1 | 0 | −0.175 | 0.811 | 0.069 | 0.330 | 0.030 | 0.470 |
| 42 | 20.019 | 18.611 | 2.250 | 1.452 | 0 | 188 | 1.007 | 26 | 0.02140 | 0.556 | 1 | 1 | −0.111 | 0.930 | 0.112 | 0.645 | 0.050 | 1.563 |
| 43 | 44.605 | 37.712 | 2.068 | 1.487 | 0 | 300 | 0.522 | 31 | 0.01076 | 0.333 | 0 | 0 | 0.631 | 0.845 | 0.046 | 0.719 | 0.012 | 0.855 |
| 44 | 18.455 | 16.839 | 0.612 | 0.453 | 1 | 183 | 1.255 | 18 | 0.01916 | 0.345 | 1 | 0 | −1.146 | 0.912 | 0.033 | 0.741 | 0.068 | 1.600 |
| 45 | 34.097 | 29.917 | 4.240 | 1.912 | 0 | 232 | 1.969 | 26 | 0.01882 | 0.489 | 1 | 1 | −0.546 | 0.877 | 0.124 | 0.451 | 0.058 | 2.458 |
| 46 | 44.115 | 41.238 | 5.668 | 3.013 | 0 | 303 | 2.014 | 51 | 0.02121 | 1.082 | 1 | 1 | −0.083 | 0.935 | 0.128 | 0.532 | 0.046 | 3.096 |
| 47 | 24.751 | 21.197 | 3.504 | 2.636 | 0 | 174 | 0.772 | 28 | 0.01932 | 0.541 | 0 | 0 | 1.324 | 0.856 | 0.142 | 0.752 | 0.031 | 1.313 |
| 48 | 3.989 | 3.485 | 0.169 | 0.127 | 1 | 51 | 0.194 | 31 | 0.01038 | 0.322 | 0 | 0 | −0.388 | 0.874 | 0.042 | 0.754 | 0.049 | 0.516 |
| 49 | 2.727 | 2.591 | 0.130 | 0.040 | 1 | 42 | 0.023 | 3 | 0.00555 | 0.017 | 0 | 0 | 0.000 | 0.950 | 0.048 | 0.306 | 0.008 | 0.040 |
| 50 | 41.849 | 36.892 | 6.298 | 2.174 | 0 | 287 | 0.654 | 20 | 0.01808 | 0.362 | 0 | 0 | 1.158 | 0.882 | 0.150 | 0.345 | 0.016 | 1.016 |
| 51 | 3.117 | 2.633 | 0.260 | 0.142 | 1 | 41 | 0.096 | 6 | 0.00275 | 0.016 | 0 | 0 | 0.030 | 0.845 | 0.083 | 0.547 | 0.031 | 0.112 |
| 52 | 40.877 | 38.435 | 2.909 | 1.178 | 0 | 270 | 1.859 | 55 | 0.00550 | 0.303 | 1 | 1 | −0.983 | 0.940 | 0.071 | 0.405 | 0.045 | 2.162 |
| 53 | 41.760 | 39.031 | 4.244 | 3.072 | 0 | 295 | 2.027 | 22 | 0.02139 | 0.471 | 0 | 0 | 0.574 | 0.935 | 0.102 | 0.724 | 0.049 | 2.498 |
| 54 | 15.011 | 12.896 | 1.683 | 1.072 | 1 | 150 | 0.995 | 27 | 0.01915 | 0.517 | 1 | 0 | −0.440 | 0.859 | 0.112 | 0.637 | 0.066 | 1.512 |
| 55 | 8.441 | 7.560 | 0.392 | 0.102 | 0 | 100 | 0.204 | 34 | 0.00457 | 0.155 | 1 | 0 | −0.258 | 0.896 | 0.046 | 0.259 | 0.024 | 0.359 |
| 56 | 7.009 | 6.042 | 0.394 | 0.266 | 1 | 83 | 0.357 | 39 | 0.00292 | 0.114 | 1 | 0 | −0.205 | 0.862 | 0.056 | 0.674 | 0.051 | 0.471 |
| 57 | 27.092 | 24.343 | 0.856 | 0.344 | 0 | 245 | 1.687 | 40 | 0.00961 | 0.384 | 1 | 1 | −1.727 | 0.899 | 0.032 | 0.402 | 0.062 | 2.071 |
| 58 | 20.371 | 17.270 | 0.672 | 0.227 | 0 | 141 | 0.641 | 25 | 0.01888 | 0.472 | 1 | 1 | −0.886 | 0.848 | 0.033 | 0.338 | 0.031 | 1.113 |
| 59 | 12.644 | 10.903 | 1.959 | 1.036 | 1 | 131 | 0.839 | 4 | 0.00674 | 0.027 | 0 | 0 | 0.170 | 0.862 | 0.155 | 0.529 | 0.066 | 0.866 |
| 60 | 25.272 | 22.153 | 2.231 | 1.456 | 0 | 180 | 0.489 | 32 | 0.01331 | 0.426 | 0 | 0 | 0.541 | 0.877 | 0.088 | 0.652 | 0.019 | 0.915 |
| 61 | 13.051 | 12.056 | 0.849 | 0.486 | 0 | 90 | 0.275 | 24 | 0.00140 | 0.034 | 0 | 0 | 0.177 | 0.924 | 0.065 | 0.573 | 0.021 | 0.309 |
| 62 | 19.520 | 17.079 | 2.800 | 2.189 | 0 | 187 | 0.695 | 20 | 0.01026 | 0.205 | 0 | 0 | 1.289 | 0.875 | 0.143 | 0.782 | 0.036 | 0.900 |
| 63 | 33.949 | 27.936 | 4.672 | 3.587 | 0 | 253 | 0.424 | 30 | 0.00585 | 0.176 | 0 | 0 | 2.988 | 0.823 | 0.138 | 0.768 | 0.012 | 0.600 |
| 64 | 11.327 | 9.101 | 1.127 | 0.430 | 1 | 121 | 0.208 | 16 | 0.01516 | 0.242 | 1 | 0 | −0.020 | 0.804 | 0.099 | 0.382 | 0.018 | 0.450 |
| 65 | 2.835 | 2.591 | 0.159 | 0.123 | 1 | 45 | 0.092 | 8 | 0.00218 | 0.017 | 0 | 0 | 0.013 | 0.914 | 0.056 | 0.775 | 0.033 | 0.109 |
| 66 | 18.235 | 16.915 | 2.843 | 1.326 | 1 | 169 | 0.483 | 53 | 0.01992 | 1.056 | 1 | 0 | −0.213 | 0.928 | 0.156 | 0.466 | 0.026 | 1.539 |
| 67 | 10.164 | 9.593 | 1.441 | 0.950 | 1 | 80 | 0.197 | 23 | 0.00795 | 0.183 | 0 | 0 | 0.570 | 0.944 | 0.142 | 0.659 | 0.019 | 0.380 |
| 68 | 25.198 | 21.991 | 3.930 | 1.173 | 0 | 223 | 1.163 | 10 | 0.00897 | 0.090 | 1 | 1 | −0.080 | 0.873 | 0.156 | 0.298 | 0.046 | 1.253 |
| 69 | 14.696 | 12.971 | 0.654 | 0.270 | 1 | 103 | 0.738 | 18 | 0.00129 | 0.023 | 1 | 0 | −0.491 | 0.883 | 0.045 | 0.413 | 0.050 | 0.761 |
| 70 | 36.096 | 29.404 | 5.140 | 1.768 | 0 | 328 | 1.799 | 69 | 0.01644 | 1.134 | 1 | 1 | −1.165 | 0.815 | 0.142 | 0.344 | 0.050 | 2.933 |
| 71 | 32.222 | 26.234 | 3.696 | 2.774 | 0 | 222 | 1.545 | 33 | 0.00876 | 0.289 | 0 | 0 | 0.939 | 0.814 | 0.115 | 0.750 | 0.048 | 1.834 |
| 72 | 4.290 | 3.888 | 0.269 | 0.133 | 0 | 60 | 0.257 | 10 | 0.02226 | 0.223 | 0 | 0 | −0.346 | 0.906 | 0.063 | 0.496 | 0.060 | 0.480 |
| 73 | 36.804 | 32.596 | 4.949 | 3.782 | 0 | 240 | 0.806 | 21 | 0.01782 | 0.374 | 0 | 0 | 2.603 | 0.886 | 0.134 | 0.764 | 0.022 | 1.180 |
| 74 | 35.046 | 29.002 | 1.254 | 0.556 | 0 | 237 | 1.147 | 49 | 0.01193 | 0.585 | 1 | 1 | −1.176 | 0.828 | 0.036 | 0.443 | 0.033 | 1.732 |
| 75 | 27.660 | 22.785 | 2.083 | 0.611 | 0 | 180 | 0.633 | 28 | 0.01393 | 0.390 | 1 | 1 | −0.412 | 0.824 | 0.075 | 0.293 | 0.023 | 1.023 |
| 76 | 2.389 | 2.091 | 0.212 | 0.153 | 0 | 35 | 0.149 | 1 | 0.00764 | 0.008 | 0 | 0 | −0.004 | 0.875 | 0.089 | 0.718 | 0.062 | 0.157 |
| 77 | 17.951 | 15.662 | 1.907 | 0.489 | 1 | 120 | 0.694 | 22 | 0.01549 | 0.341 | 1 | 0 | −0.546 | 0.872 | 0.106 | 0.256 | 0.039 | 1.035 |
| 78 | 32.903 | 27.961 | 4.656 | 1.413 | 0 | 228 | 0.408 | 62 | 0.01463 | 0.907 | 0 | 0 | 0.098 | 0.850 | 0.142 | 0.304 | 0.012 | 1.315 |
| 79 | 26.000 | 20.962 | 4.012 | 1.177 | 0 | 175 | 0.290 | 52 | 0.00340 | 0.177 | 0 | 0 | 0.711 | 0.806 | 0.154 | 0.293 | 0.011 | 0.467 |
| 80 | 12.306 | 10.488 | 0.484 | 0.216 | 1 | 126 | 0.278 | 15 | 0.01670 | 0.251 | 1 | 0 | −0.312 | 0.852 | 0.039 | 0.446 | 0.023 | 0.529 |
| 81 | 3.855 | 3.527 | 0.339 | 0.177 | 1 | 57 | 0.037 | 39 | 0.00181 | 0.070 | 0 | 0 | 0.070 | 0.915 | 0.088 | 0.521 | 0.010 | 0.107 |
| 82 | 39.651 | 34.024 | 5.067 | 1.627 | 0 | 330 | 1.630 | 57 | 0.00167 | 0.095 | 1 | 1 | −0.098 | 0.858 | 0.128 | 0.321 | 0.041 | 1.725 |
| 83 | 22.357 | 19.500 | 3.249 | 2.014 | 0 | 150 | 0.856 | 58 | 0.00373 | 0.216 | 0 | 0 | 0.942 | 0.872 | 0.145 | 0.620 | 0.038 | 1.072 |
| 84 | 35.344 | 31.980 | 5.019 | 2.798 | 0 | 320 | 1.411 | 108 | 0.00750 | 0.810 | 0 | 0 | 0.577 | 0.905 | 0.142 | 0.557 | 0.040 | 2.221 |

**Table A1.** *Cont.*

| i | v | c | p | r | k | t | a | d | u | q | s | w | e | c/v | p/v | r/p | a/v | a+q |
|---|---|---|---|---|---|---|---|---|---|---|---|---|---|---|---|---|---|---|
| 85 | 21.776 | 19.538 | 1.951 | 1.325 | 0 | 150 | 1.420 | 20 | 0.01857 | 0.371 | 1 | 1 | −0.466 | 0.897 | 0.090 | 0.679 | 0.065 | 1.791 |
| 86 | 34.789 | 32.232 | 3.977 | 2.627 | 0 | 299 | 0.577 | 108 | 0.00973 | 1.050 | 0 | 0 | 1.000 | 0.926 | 0.114 | 0.661 | 0.017 | 1.627 |
| 87 | 24.592 | 20.044 | 2.376 | 1.808 | 0 | 226 | 0.965 | 33 | 0.00790 | 0.261 | 0 | 0 | 0.583 | 0.815 | 0.097 | 0.761 | 0.039 | 1.226 |
| 88 | 15.851 | 14.087 | 1.372 | 0.789 | 0 | 115 | 0.924 | 15 | 0.01004 | 0.151 | 1 | 0 | −0.285 | 0.889 | 0.087 | 0.575 | 0.058 | 1.075 |
| 89 | 24.813 | 22.463 | 0.985 | 0.414 | 0 | 221 | 1.690 | 61 | 0.01850 | 1.129 | 1 | 1 | −2.405 | 0.905 | 0.040 | 0.420 | 0.068 | 2.819 |
| 90 | 37.965 | 31.577 | 5.782 | 2.253 | 0 | 250 | 1.282 | 49 | 0.01144 | 0.561 | 0 | 0 | 0.410 | 0.832 | 0.152 | 0.390 | 0.034 | 1.843 |
| 91 | 36.238 | 30.127 | 1.810 | 0.710 | 0 | 313 | 0.645 | 32 | 0.00692 | 0.222 | 1 | 1 | −0.157 | 0.831 | 0.050 | 0.392 | 0.018 | 0.867 |
| 92 | 24.374 | 20.770 | 3.242 | 1.353 | 0 | 164 | 1.632 | 13 | 0.00750 | 0.098 | 1 | 1 | −0.376 | 0.852 | 0.133 | 0.417 | 0.067 | 1.730 |
| 93 | 38.579 | 35.418 | 1.738 | 0.538 | 0 | 254 | 1.889 | 45 | 0.00567 | 0.255 | 1 | 1 | −1.606 | 0.918 | 0.045 | 0.310 | 0.049 | 2.144 |
| 94 | 32.091 | 30.334 | 4.360 | 3.047 | 0 | 227 | 1.611 | 24 | 0.01391 | 0.334 | 0 | 0 | 1.103 | 0.945 | 0.136 | 0.699 | 0.050 | 1.945 |
| 95 | 37.075 | 30.196 | 4.791 | 1.250 | 0 | 251 | 1.031 | 67 | 0.00586 | 0.392 | 1 | 1 | −0.174 | 0.814 | 0.129 | 0.261 | 0.028 | 1.423 |
| 96 | 18.902 | 15.202 | 0.884 | 0.572 | 0 | 130 | 0.264 | 33 | 0.01016 | 0.335 | 1 | 1 | −0.027 | 0.804 | 0.047 | 0.647 | 0.014 | 0.599 |
| 97 | 3.752 | 3.587 | 0.144 | 0.045 | 1 | 50 | 0.255 | 7 | 0.02054 | 0.144 | 0 | 0 | −0.353 | 0.956 | 0.038 | 0.315 | 0.068 | 0.399 |
| 98 | 2.736 | 2.299 | 0.209 | 0.082 | 0 | 40 | 0.079 | 9 | 0.02206 | 0.199 | 0 | 0 | −0.195 | 0.840 | 0.076 | 0.391 | 0.029 | 0.278 |
| 99 | 37.163 | 30.973 | 4.050 | 1.200 | 0 | 250 | 2.079 | 79 | 0.00865 | 0.683 | 1 | 1 | −1.563 | 0.833 | 0.109 | 0.296 | 0.056 | 2.762 |
| 100 | 5.889 | 4.852 | 0.420 | 0.237 | 0 | 76 | 0.055 | 23 | 0.01301 | 0.299 | 0 | 0 | −0.117 | 0.824 | 0.071 | 0.565 | 0.009 | 0.354 |

**Appendix B**

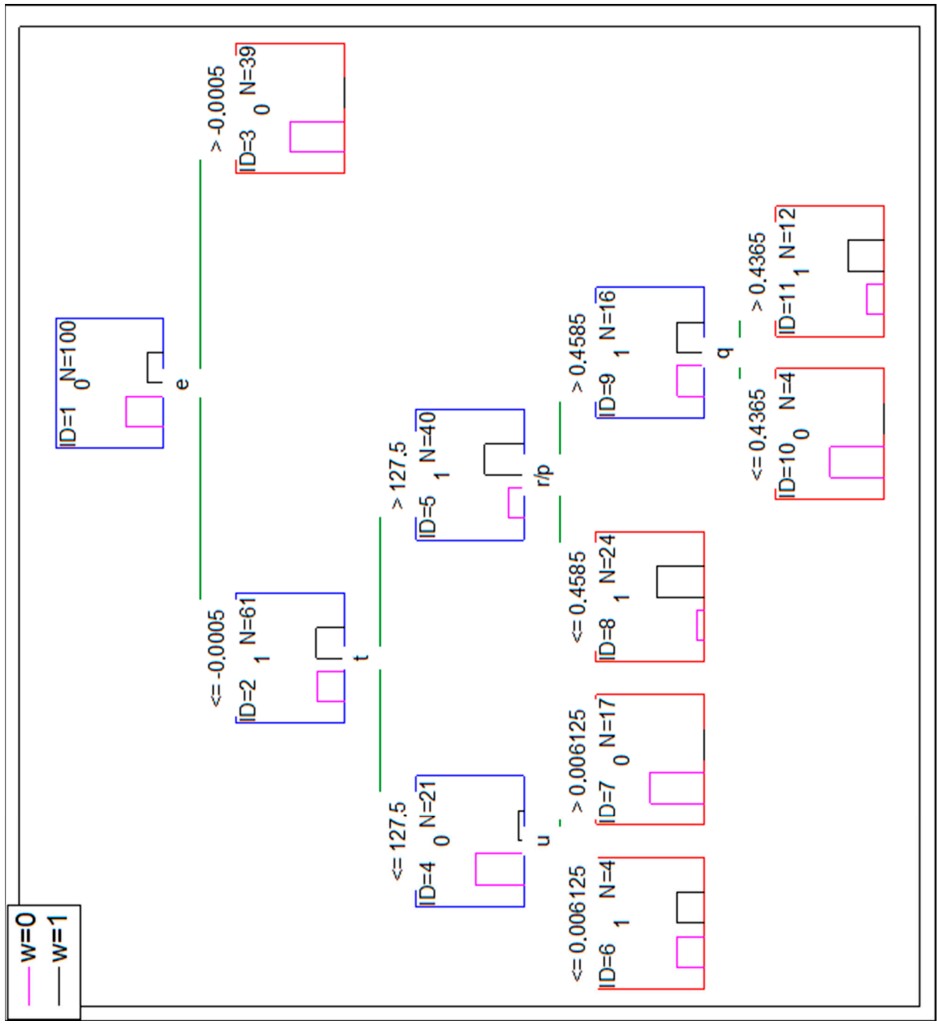

**Figure A1.** The decision tree with the extended input for mod-1.

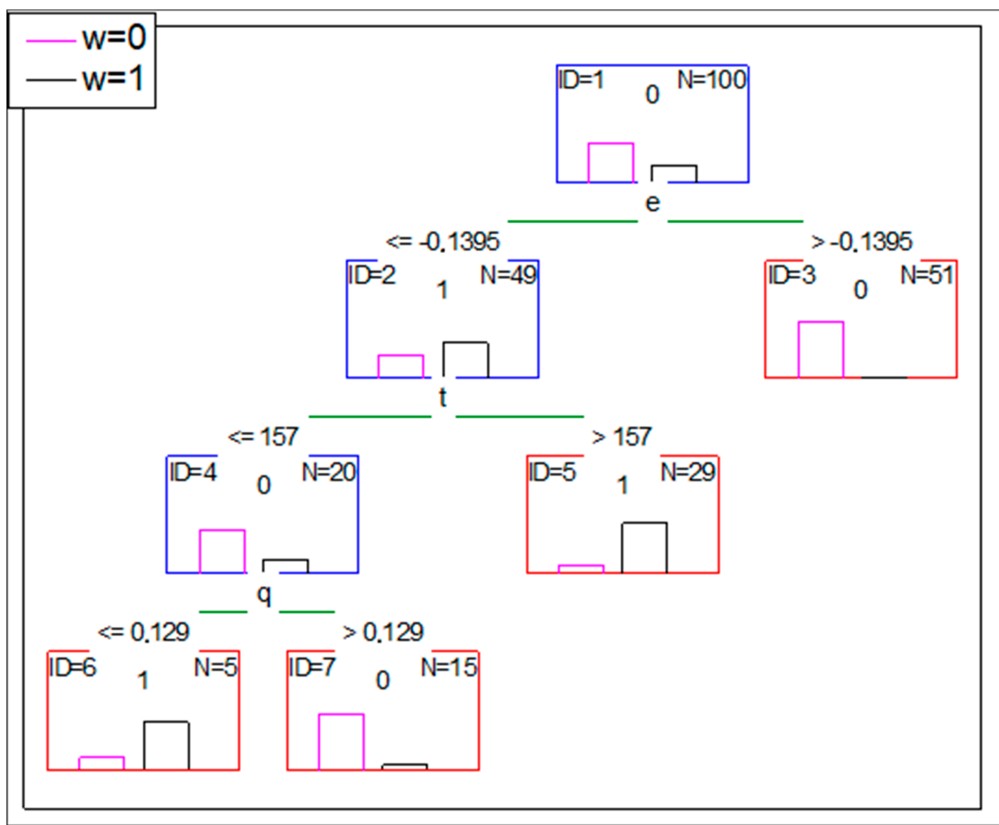

**Figure A2.** The decision tree for mod-2.

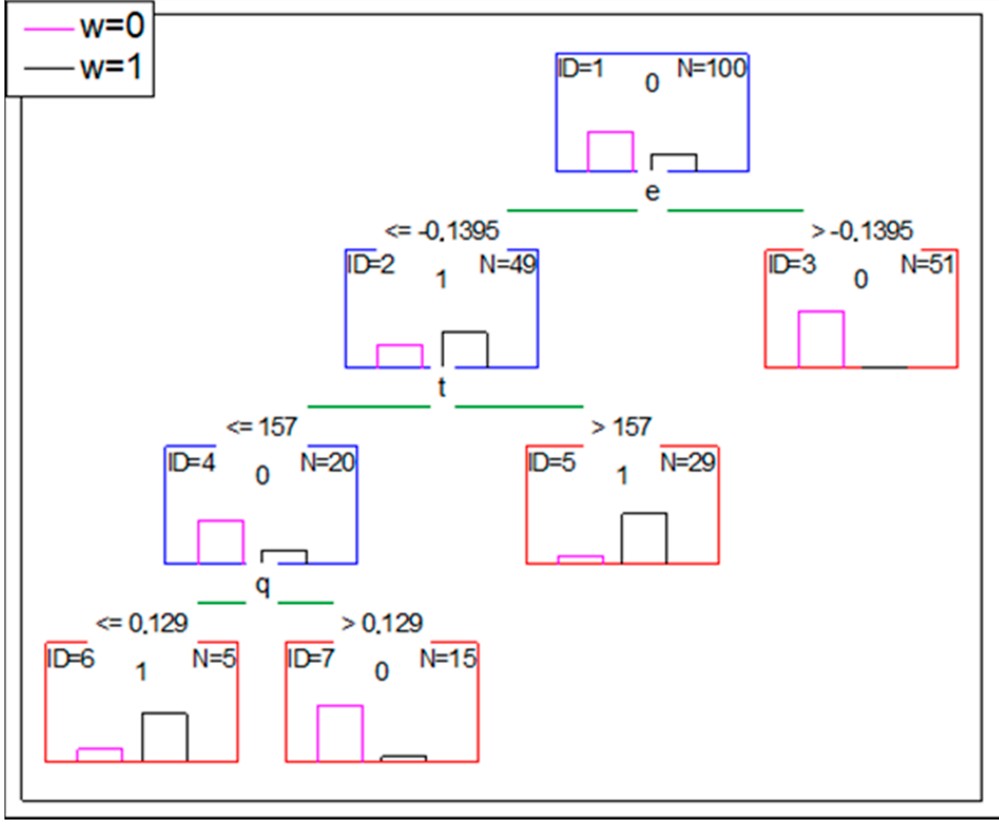

**Figure A3.** The decision tree for mod-2 (based on the extended input).

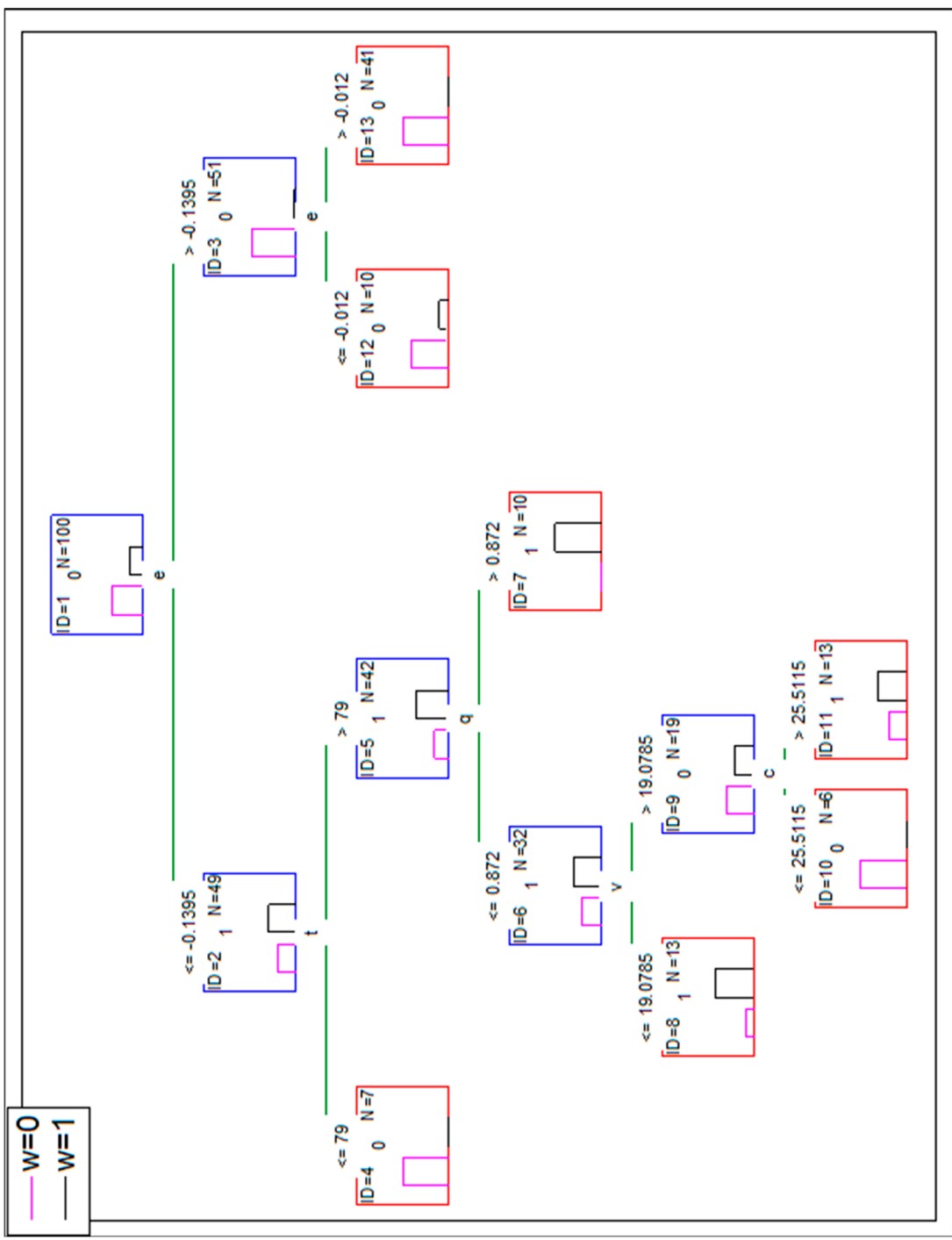

**Figure A4.** The decision tree for mod-3.

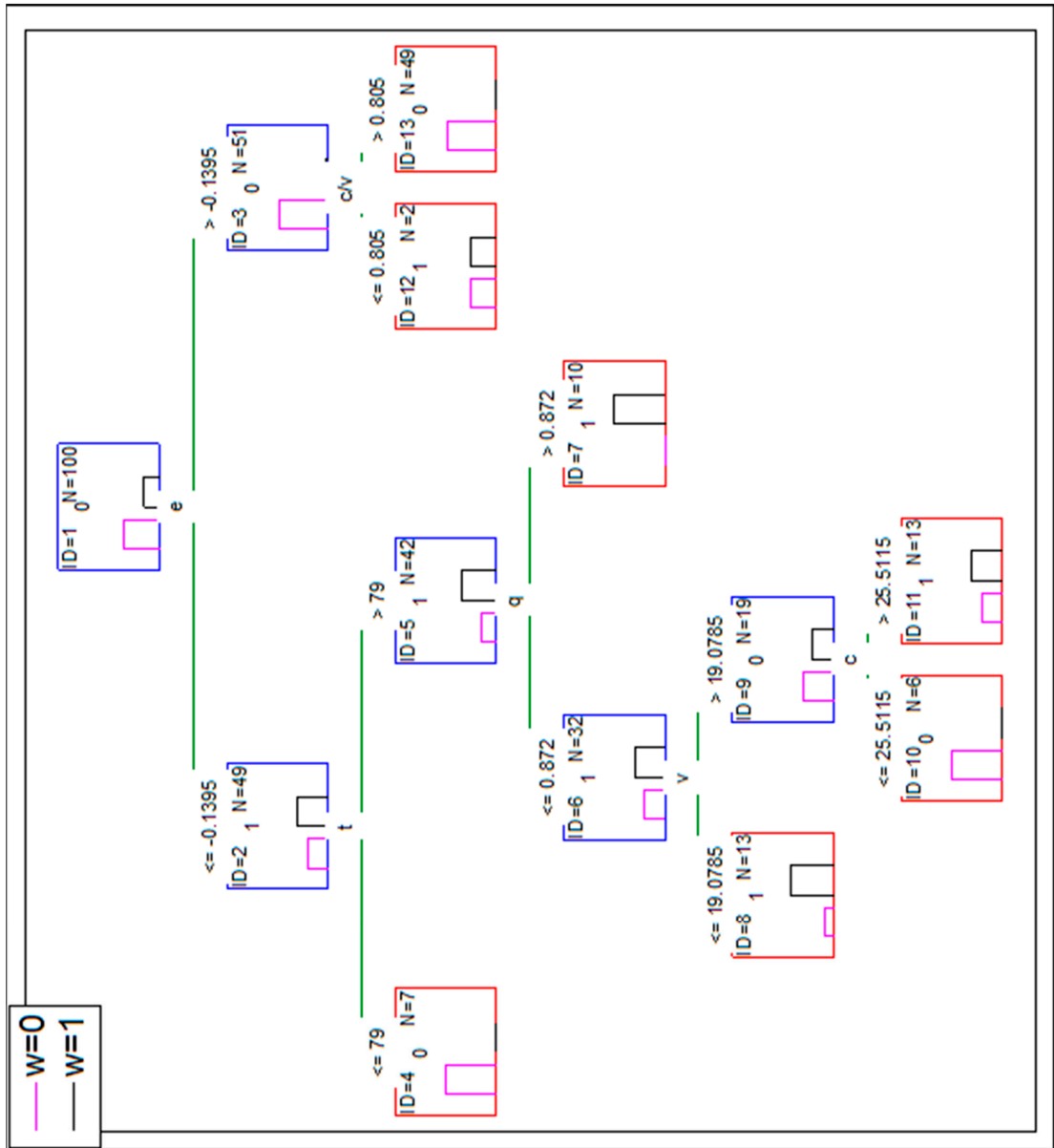

**Figure A5.** The decision tree for mod-3 (based on the extended input).

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
