# Peer review of "Quantitative Risk Assessment in Construction Disputes Based on Machine Learning Tools"

_symmetry, doi:10.3390/sym13050744_

Round 1

Reviewer 1 Report

Thank you for the opportunity to review your paper. 

As you stated, the main goal of your paper is to find an optimal strategy for construction general contractors to address contract disputes. I have two major concerns and a few minor revisions. 

First, 

Lines 162-164 – the authors claim that there use of machine learning techniques, and, specifically ANN and DT, have not been previously used in construction; however, several papers have been previously published in this area dating back to at least 2005. Therefore, the method is not novel. Here are a few examples:

https://ascelibrary.org/doi/abs/10.1061/(ASCE)0887-3801(2010)24:1(73)
https://ascelibrary.org/doi/abs/10.1061/(ASCE)0887-3801(2005)19:4(387)
https://www.sciencedirect.com/science/article/abs/pii/S0263786315001416
https://ascelibrary.org/doi/abs/10.1061/%28ASCE%29CO.1943-7862.0002027
https://ascelibrary.org/doi/abs/10.1061/(ASCE)CP.1943-5487.0000148
https://www.tandfonline.com/doi/abs/10.3846/13923730.2013.768544

Second, while I am not an ANN expert, I am concerned about the extremely small sample size used to train your model and the use of a simulated dataset based on the rules of your original 10 contracts to test your model. If you are using the same rules from your original set (where you trained your ANN) to create a simulated dataset, won't you automatically expect high success rates? 

Additionally, you also admit in your conclusions that "it cannot be stated with a high certainty that the proposed machine learning tools can effectively help a GC with the decisions concerning dispute resolution." If the method is not novel, as shown above, and you admit that there is not a high level of confidence in achieving your primary goal, I question the value of the paper. 

Other, minor comments: 

Line 11 – hyphenate “one time” and “single product”

I do not believe that sufficient literature review has been conducted for the authors to claim that they have identified the most common causes of conflicts between parties when they primarily relied upon one Polish survey study.

Line 307 – use a semicolon before “however” at the end of the line

Multiple lines – delete the space between a number and %, e.g., Line 379 “Rule 4 is 100 %”

Line 412 – Replace “which” with “that”

Line 484 – Figure 3 and others created using the same software are far too low resolution

Line 485 – use a semicolon before “however”

Line 617, 651 – avoid contractions

Author Response

We are very grateful for the constructive comments and thoughtful suggestions. We made substantial changes to the manuscript in accordance with these proposals, therefore we believe that the quality of the manuscript improved considerably as a result.

Revisions in the text of the manuscript are highlighted using the "Track Changes" function. Our answers to the questions and issues raised are written in italic.

Reviewer 2 Report

Thank you for the opportunity of reviewing your manuscript. It addresses a topic with potential significant implications in business. The authors use an interesting method and present valuable information. The methodology is appropriate and the results are adequately discussed. I suggest authors to include a final section to discuss theoretical and particularly practical implications of their findings, and to add limitations and further suggestions for reserach. Moreover, a style revision is suggested, including proofreading by a specialist / native. Good luck!

Author Response

We are very grateful for the constructive comments and thoughtful suggestions.

Revisions in the text of the manuscript are highlighted using the "Track Changes" function. Our answers to the questions and issues raised are written in italic.

Reviewer 3 Report

I think the two parts to be revised are the introduction and the conclusions. In the introduction, the word "conclusions" should be removed. Placed in the middle of the introduction it confuses the reader. The introduction should, therefore, be restructured.

In the conclusion of the paper, the implications of the model should be added. That is, what use can be made of the model, what are the policies? The authors are much more focused on the (mathematical) application of the model than on its actual and effective use. Those parts should be revised then.

Author Response

(The authors gave the same response as above.)

Reviewer 4 Report

I consider the article to be an interesting study, which brings the novelty of machine learning tools (decision trees and artificial neural networks) in such an actual field of construction (or at least in full swing before the COVID pandemic). 
As a veteran in data mining experiments, I appreciate authors’ concentration in other fields of activity, and I appreciate their inclination towards interdisciplinarity. The predictions of machine learning tools can bring support even in the case of relations between entrepreneurs and customers (an interesting way of approaching), and in the case of avoiding conflict situations among them.  I appreciate the results as being original (Turnitin similarity index is only 5%, which is very good).

I also appreciate that the authors used an advanced analytics software package StatSoft (now Dell's) Statistics to perform the experiments, maybe for the future they could also lean towards specific Machine Learning packages from R (eg, dplyr, ggplot2, mlr3, xgboost , cart, etc.) or even Python which can offer other functionalities. 

There are some issues that I would recommend to the authors to review.
I noticed a possible spelling error from the abstract, namely the word "Sueing". If it is "suing" (the correct form of the verb "to sue", and not the American basketball player “Justice Sueing”) then I recommend the authors to correct everywhere in the text. If it is another word specific to the field of constructions (?) then I recommend the authors to use a synonym, or to explain it. There is also another word, "concequent" on page 6 just after the 2nd form. I believe it to be "consequent".
I advise the authors to try and decide whether they use British or American English. Although they have typical words spelled for British like “analyse”, they also use the Am.Eng. “Analyzed” and “analyze” just before and after Table 17, on page 18 etc.
The same thing is also with the word (non-standard) “Similarily” on page 20, 2nd paragraph. 

I am also wondering what decimal symbol the authors finally use, because as I can distinguish from the paper, they use both the decimal point (specific to UK, US etc.) in Table 1, but also the decimal comma (specific to Poland, Hungary etc.) in the next table (Table 2), and again the decimal point in Table 3 etc. Please decide what you use. 
I also tried to distinguish what was the decimal symbol used in the Statistica's generated figures, but their resolution is low.  I appreciate the Statistica software, but I believe the resolutions of the figures generated to be quite low. Perhaps an enhancement of the figures no. 3, 4 and 6 would be advised, for a better graphical resolution and of course to be better understood and read. 

Another weaker point of the study is that the set of data provided by the construction company for this article is narrow i.e. 10 real projects described by several time and cost parameters, fact recognized even by the authors. Even if an accuracy of over 93% is achieved, I strongly believe that a larger database would have been better for the final decision making process. 

I don't consider the spelling, resolution, decimal symbol errors to be extremely critical, because there are things that probably happen when several authors with different styles work on a study / article, but I do recommend a clear unitary final reading and a final verification from a clear mind. :) 
The work and the experiment underlying it seem to have been really worked on, but the final appearance gives the impression to have been made in a hurry, without further verification. Perhaps a more careful reading by a native English speaker would be required. 
“Co nagle to po diable” an old Polish proverb would say (approx. Eng. “Haste makes waste”). 

Author Response

We are very grateful for the constructive comments and thoughtful suggestions. We made some changes to the manuscript in accordance with these proposals, therefore we believe that the quality of the manuscript improved considerably as a result.

 Revisions in the text of the manuscript are highlighted using the "Track Changes" function. Our answers to the questions and issues raised are written in italic.

Reviewer 5 Report

Thank you very much for the opportunity to revise this article. The subject of the article is the quantitative assessment of risks in construction disputes based on machine learning tools. The basis is an analysis of the most common causes of conflicts between the parties to the construction contract. Using DT and ANN, the authors present the application possibilities of tools supporting the decision-making process of the supplier in a conflict situation with the client.

I have the following comments on the article: on line 132 you have a paragraph called Conclusion. I recommend considering a change in the Conclusion or moving this area to the conclusion for an overall assessment - the final conclusion.

The article is too long, I recommend joining the tables, e.g. Table 9, 10,11 (mod - 1) with the header corresponding to the given items. Similarly for mod-2 Tab. 12 and 13 and mod-3 Tab. 14, 15, and 16.

As the article is entitled "quantitative risk assessment", the consequences of construction disputes can be of different levels and of different types. Is it possible to extend the proposed methodology with these consequences in order to be able to express the risk more appropriately? How to deal with uncertainty in risk assessment. 

Author Response

(The authors gave the same response as above.)

Round 2

Reviewer 1 Report

Dear authors, 

Thank you for your thoughtful responses. While I question that there is real novelty in only analyzing data from one contractor, I do appreciate that you removed the claim that your approach was novel. 

I also believe it was a good idea to include a few of the papers I sent you. I think you now more actually describe your paper's place in this field. 

The only grammatical issue I still have is that you incorrectly updated line 11. I was saying to hyphenate the terms because they were acting together as a single adjective. The line should read like this: 

A construction site is a unique, one-time, and single-product factory with many parties involved and dependent on each other.

Author Response

Please find enclosed.

Reviewer 4 Report

The authors improved their article and added several corrections as suggested. The database dimension still remains low, but I understood the authors’ explanation to that.

Author Response

Thank you very much for the positive review of our article. We are glad you considered the content of the article interesting and worth of publication. Once again, we are very grateful for all the comments and suggestions, which helped us to improve the quality of the manuscript.